# Genetic correlations reveal the shared genetic architecture of transcription in human peripheral blood

Samuel W. Lukowski [1], Luke R. Lloyd-Jones[1,2], Alexander Holloway[2], Holger Kirsten[3,4], Gibran Hemani [5,6], Jian Yang [1,2], Kerrin Small [7], Jing Zhao[8], Andres Metspalu[9], Emmanouil T. Dermitzakis[10], Greg Gibson [8], Timothy D. Spector[7], Joachim Thiery[4,11], Markus Scholz [3,4], Grant W. Montgomery[1,12], Tonu Esko [9], Peter M. Visscher [1,2] & Joseph E. Powell[1,2]

Transcript co-expression is regulated by a combination of shared genetic and environmental factors. Here, we estimate the proportion of co-expression that is due to shared genetic variance. To do so, we estimated the genetic correlations between each pairwise combination of 2469 transcripts that are highly heritable and expressed in whole blood in 1748 unrelated individuals of European ancestry. We identify 556 pairs with a significant genetic correlation of which 77% are located on different chromosomes, and report 934 expression quantitative trait loci, identified in an independent cohort, with significant effects on both transcripts in a genetically correlated pair. We show significant enrichment for transcription factor control and physical proximity through chromatin interactions as possible mechanisms of shared genetic control. Finally, we construct networks of interconnected transcripts and identify their underlying biological functions. Using genetic correlations to investigate transcriptional co-regulation provides valuable insight into the nature of the underlying genetic architecture of gene regulation.

[1] Institute for Molecular Bioscience, University of Queensland, Brisbane, QLD 4072, Australia. [2] Centre for Neurogenetics and Statistical Genomics, Queensland Brain Institute, University of Queensland, Brisbane, QLD 4072, Australia. [3] Institute for Medical Informatics, Statistics and Epidemiology, University of Leipzig, Leipzig 04107, Germany. [4] LIFE Leipzig Research Center for Civilization Diseases, University of Leipzig, Leipzig 04103, Germany. [5] MRC Integrative Epidemiology Unit (IEU) at the University of Bristol, Bristol BS8 2BN, UK. [6] School of Social and Community Medicine, University of Bristol, Bristol BS8 2BN, UK. [7] Department of Twin Research and Genetic Epidemiology, King's College London, London SE1 7EH, UK. [8] School of Biology and Center for Integrative Genomics, Georgia Institute of Technology, Atlanta, GA 30332, USA. [9] Estonian Genome Center, University of Tartu, Tartu 51010, Estonia. [10] Department of Genetic Medicine and Development, University of Geneva, Geneva CH-1211, Switzerland. [11] Institute of Laboratory Medicine, Clinical Chemistry and Molecular Diagnostics, University of Leipzig, Leipzig 04103, Germany. [12] QIMR Berghofer Medical Research Institute, 300 Herston Road, Brisbane, QLD 4006, Australia. Correspondence and requests for materials should be addressed to J.E.P. (email: joseph.powell@uq.edu.au)

The co-expression of a pair of transcripts, which can be represented by their phenotypic correlation, is due to the combined influence of shared genetic and environmental factors[1]. The degree of genetic variance that is shared between a pair of transcripts is the genetic correlation, a quantity that includes the effects of all variants, including those with small effects that are not identifiable through current expression quantitative trait loci (eQTL) studies. Genetic correlations are not the same as heritability, as they represent the proportion of overlapping genetic effects, and not their absolute magnitude; the expression levels of two transcripts could both be highly heritable but not be genetically correlated, or have low heritability and be completely correlated. Genetic and environmental correlations have traditionally been identified using twin models[2], although more recently, approaches using genotype data from unrelated individuals have been developed. Mixed-model methods such as bivariate genome-wide relatedness restricted maximum likelihood (GREML)[3], can be used to estimate genetic and environmental correlations from population-level data, by providing an estimate of the genetic co-variance for two traits, such as the expression levels of a pair of transcripts. Other methods such as cross-trait linkage disequilibrium (LD) Score Regression[4] are useful alternatives when only summary statistics on genetic effects are available, although this is less powerful than using individual-level data[5].

In humans, methods utilising genome-wide genotype data from unrelated individuals have been used to estimate the total shared genetic overlap between human diseases and complex traits (i.e. pleiotropy)[3, 4, 6–8]. These studies have revealed some interesting results relating to epidemiological observations of shared co-morbidity. For example, they have confirmed and added resolution to previously known relationships, such as shared genetic control among metabolic diseases[6, 7], and identified novel relationships such as positive genetic correlations between anorexia nervosa and schizophrenia[4]. Importantly, such experiments can reveal pairs of diseases exhibiting genetic correlations close to zero, indicating that any observed co-morbidity is likely due to shared environmental factors.

Currently little is known about the total degree of shared genetic effects for expression levels of transcripts. However, there has been considerable effort to identify specific loci, which often have effects on two or more transcripts. These loci (eQTL)[9] are typically identified using traditional univariate mapping strategies[10, 11], although multivariate approaches have also been used[12, 13]. Most eQTL studies have mapped cis-acting loci only-that is loci that are located in close proximity to the transcript(s), and many incidences of cis-effects shared between transcripts have been identified[14, 15]. Recently Westra et al.[16] used a combination of cis-mapping and trans-mapping to identify 103 independent eQTL with effects on two or more transcripts. Some of these eQTLs affect multiple transcripts in trans, and support previous work showing that a cis-acting eQTL for the transcription factor KLF14 gene acts as a master trans regulator of adipose gene expression[17].

By establishing the total degree of shared genetic co-regulation between transcripts, as opposed to specific loci, we are able to: (i) identify networks consisting of transcripts located throughout the genome, whose expression levels are influenced by the same genetic variants, (ii) determine the proportion of variants on the same, or different, chromosomes and (iii) estimate the contribution of shared environmental factors to the co-variance of transcript expression.

To provide an illustrative example of how molecular processes could underlie an observed genetic correlation for pairs of transcripts, consider the following simplified scenario: a gene encodes a transcription factor which has a large number of target genes. Suppose that the variation in the transcription factor's expression levels is entirely due to a single-genetic variant. Under such a scenario, we would expect that the expression levels of the target genes would also be affected by the genetic variant. Thus, we should observe non-zero estimates of the genetic correlation between the expression levels of the transcription factor gene, and each of its target genes. A positive genetic correlation between two transcripts implies that, on average, shared genetic variants have an allelic effect in the same direction. Conversely, a negative genetic correlation implies that, on average, the allelic effect of the genetic variants is in the opposite direction.

Here, we have performed a systematic investigation of the degree of shared genetic control between pairs of expressed and highly heritable transcripts in whole blood. We have sought to understand the molecular processes that may give rise to the statistical observations of genetic correlations. We have estimated the genetic correlations between each pairwise combination of 2469 transcripts that are both highly heritable and expressed in whole blood in a cohort of 1748 unrelated individuals of European ancestry. We identified 556 pairs with a significant genetic correlation at a Bonferroni study-wide threshold ($p < 1.81 \times 10^{-8}$), of which 77% were located on different chromosomes. Using eQTL data from an independent cohort ($n = 2104$), we identified 934 incidences of eQTL with significant (Bonferroni, $p < 4.1 \times 10^{-8}$) effects on both transcripts in a genetically correlated pair. We then investigated the possible mechanisms that may lead to shared genetic control between a pair of transcripts. Our findings reveal significant enrichment for both of transcription factor control (hypergeometric test, $p = 1.14 \times 10^{-3}$) and physical proximity through chromatin interactions (Hi-C; two-tailed permutation, $p < 0.001$). Finally, we used estimates of genetic correlations to construct graph networks of interconnected transcripts and identified the biological functions underlying many of the networks. Our findings demonstrate the utility of using genetic correlations to investigate transcriptional co-regulation and to gain valuable insight into the nature of the underlying genetic architecture of gene regulation. All results

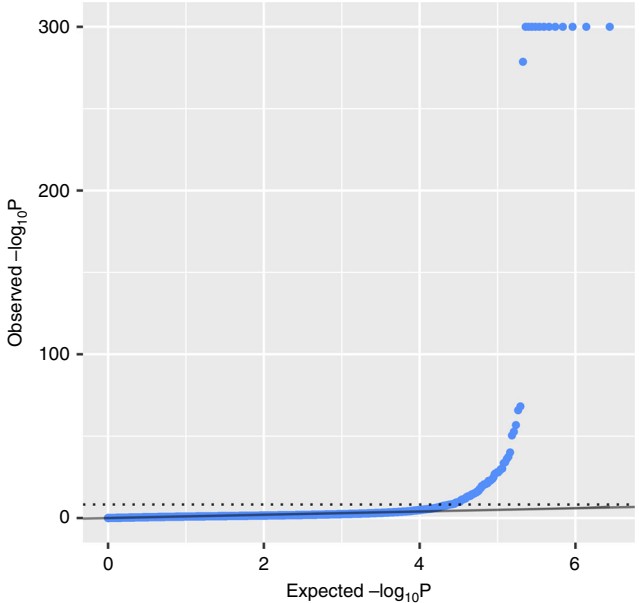

**Fig. 1** A quantile–quantile plot shows the observed (y-axis, −log10 scale) and expected p-values (x-axis, −log10 scale) for each transcript pair in the analysis (2 755 498) as determined by a $\chi^2$ test. The dotted line marks the multiple testing threshold (Bonferroni: $p < 1.81 \times 10^{-8}$), which includes 556 transcript pairs. Of these significant transcript pairs, 77.0% are located on different chromosomes

**Table 1 Summary of GCTA results for 20 significant interchromosomal $\hat{r}_G$ pairs**

| Transcript 1 | Transcript 2 | Gene 1 | Gene 2 | $r_P$ | $\hat{r}_G$ | $\hat{r}_G$SE | Transcript 1 $h^2$ | Transcript 2 $h^2$ | Transcript 1 chr | Transcript 2 chr | P |
|---|---|---|---|---|---|---|---|---|---|---|---|
| ILMN_1737195 | ILMN_1795317 | CENPK | SCAND1 | −0.008 | −0.697 | 0.054 | 0.999 | 0.4 | 5 | 20 | 40.39 |
| ILMN_1787410 | ILMN_1737195 | EIF6 | CENPK | 0 | −0.683 | 0.056 | 0.5 | 0.999 | 20 | 5 | 36.49 |
| ILMN_2313901 | ILMN_1767481 | PAM | XRCC6BP1 | −0.017 | −0.67 | 0.058 | 0.999 | 0.832 | 5 | 12 | 33.14 |
| ILMN_1704985 | ILMN_1705746 | CYP27A1 | LSG1 | −0.001 | −0.655 | 0.059 | 0.999 | 0.435 | 2 | 3 | 30.91 |
| ILMN_1797764 | ILMN_1737195 | RPL22L1 | CENPK | −0.03 | −0.649 | 0.062 | 0.578 | 0.999 | 3 | 5 | 27.92 |
| ILMN_2313901 | ILMN_1718668 | PAM | ZBTB26 | 0.003 | −0.635 | 0.062 | 0.999 | 0.376 | 5 | 9 | 26.89 |
| ILMN_1699071 | ILMN_1676528 | C21ORF7 | BTN3A2 | −0.021 | −0.609 | 0.064 | 0.53 | 0.999 | 21 | 6 | 23.74 |
| ILMN_1695092 | ILMN_1676528 | WRB | BTN3A2 | −0.079 | −0.606 | 0.065 | 0.592 | 0.999 | 21 | 6 | 22.95 |
| ILMN_1743145 | ILMN_1786050 | ERAP2 | RBBP9 | −0.001 | −0.618 | 0.067 | 0.999 | 0.452 | 5 | 20 | 22.54 |
| ILMN_1676528 | ILMN_1676528 | STAT4 | BTN3A2 | 0.153 | −0.557 | 0.068 | 0.377 | 0.999 | 2 | 6 | 18.59 |
| ILMN_1737195 | ILMN_1793770 | CENPK | DNAJB6 | −0.017 | 0.628 | 0.066 | 0.999 | 0.59 | 5 | 7 | 23.74 |
| ILMN_1806056 | ILMN_2102721 | CEACAM8 | DEFA1B | 0.884 | 0.901 | 0.094 | 0.252 | 0.286 | 19 | 8 | 24.03 |
| ILMN_1659227 | ILMN_1691071 | CD79A | FCRLA | 0.773 | 0.935 | 0.097 | 0.315 | 0.394 | 19 | 1 | 24.26 |
| ILMN_1806056 | ILMN_1725661 | CEACAM8 | DEFA1B | 0.886 | 0.908 | 0.091 | 0.252 | 0.275 | 19 | 8 | 25.72 |
| ILMN_2095653 | ILMN_1815205 | AFMID | LYZ | 0.565 | 1 | 0.1 | 0.667 | 0.921 | 17 | 12 | 25.82 |
| ILMN_2055781 | ILMN_1701237 | KLRF1 | SH2D1B | 0.7 | 0.923 | 0.091 | 0.507 | 0.551 | 12 | 1 | 26.45 |
| ILMN_1757636 | ILMN_2391512 | C5ORF35 | NAAA | 0.082 | 0.666 | 0.06 | 0.999 | 0.672 | 5 | 4 | 30.9 |
| ILMN_1743145 | ILMN_1752932 | ERAP2 | MPZL2 | 0.045 | 0.677 | 0.06 | 0.999 | 0.809 | 5 | 11 | 31.8 |
| ILMN_2366212 | ILMN_1691071 | CD79B | FCRLA | 0.842 | 0.963 | 0.072 | 0.281 | 0.394 | 17 | 1 | 43.07 |
| ILMN_2095653 | ILMN_2162972 | AFMID | LYZ | 0.773 | 0.965 | 0.056 | 0.667 | 0.911 | 17 | 12 | 68.82 |

*P-values are shown as −log10. Transcripts are identified by the Illumina HT12-V4.0 array probe ID and are separated into negative $\hat{r}_G$ pairs (upper) and positive $\hat{r}_G$ pairs (lower).*

are made publicly available at http://computationalgenomics.com.au/shiny/rg/.

## Results

**Estimates of genetic correlations between transcript pairs**. We used a bivariate GREML model[3], implemented through GCTA software[18], to estimate the genetic correlation ($r_G$) for the expression levels of pairs of transcripts. Expression levels were assayed using Illumina HT12-v4.0 expression arrays. We restricted our analysis to 2469 transcripts whose expression levels are highly heritable ($h_g^2 > 0.25$), and estimated $r_G$ for each pairwise combination.

We obtained converged results for 2 755 498 probe pairs, representing 90.5% of the 3 045 512 we tested. Of the 290 014 non-converging pairs, 96% were from probe pairs with a phenotypic correlation ($\hat{r}_P$) of 0, and as such, we would not expect high $\pm r_G$[1]. We identified 556 pairs of probes significant at a study-wide Bonferroni threshold of $p < 1.81 \times 10^{-8}$ (0.05/2755498); Fig. 1, Supplementary Data 1. Of the 556, 91% (506) of probes pairs map to different RefSeq genes, with 428 (77%) of pairs containing transcripts located on different chromosomes, and 128 (23%) on the same chromosome (hereon-termed interchromosomal and intrachromosomal, respectively). Pairs of transcripts that map to the same gene typically represent alternate isoforms, and thus intrachromosomal $\hat{r}_G$ should be interpreted with caution; although the identification of non-shared genetic control of transcript isoforms is of interest. The significant interchromosomal pairs are of particular relevance, as they implicitly demonstrate that the majority of shared genetic loci will be located on a different chromosome to at least one of the transcripts. This observation is supported by evidence showing that, on average, the majority of genetic variance for gene expression is located on different chromosomes to the location of the transcript[19]. For the 428 significant interchromosomal pairs, 36% (153) have a negative $\hat{r}_G$ (mean=−0.65), and thus 64% (275) have a positive $\hat{r}_G$ (mean = 0.77). A summary of the ten most significant positive and negative is given in Table 1. Given the high correlation structure among transcripts, and thus the conservative nature of the Bonferroni adjustment, we calculated the study-wide false discovery rate (FDR) to identify transcript pairs for subsequent functional analyses. We identified 14 991 pairs at a study-wide FDR threshold of 0.05, of

which 7886 have a positive $\hat{r}_G$ and 7105 have a negative $\hat{r}_G$, with the mean absolute $\hat{r}_G$ of 0.71, and mean SE of 0.31 (Supplementary Data 2). Of the 14 991 pairs, 971 (6.5%) are intrachromosomal and 14 020 (93.5%) are interchromosomal. The high percentage of pairs of transcripts that lack a significant $\hat{r}_G$ is expected because the majority have an $\hat{r}_P$ close to zero, with 96% between −0.1 and 0.1 (Supplementary Fig. 1A), implying little evidence of shared genetic control. All $\hat{r}_G$ results are publicly available to browse and download at http://computationalgenomics.com.au/shiny/rg/.

We observe a clear relationship in the sign direction of the phenotypic and genetic correlations (Supplementary Fig. 2), suggesting that the shared environmental effects are typically in the same direction. Of the 428 *trans* pairs significant at the Bonferroni threshold of $p < 1.81 \times 10^{-8}$, there were 81.8% of transcript pairs with the same sign of $\hat{r}_P$ and $\hat{r}_G$ direction ($R^2 = 0.48$; Supplementary Fig. 2). Of the remaining 18.2%, 36.6% have an $|\hat{r}_P| \leq 0.01$, and thus, may be incorrectly signed due to sampling variance of the correlation estimates. Closer inspection of transcript pairs with a threshold of $\hat{r}_P \geq 0.2$ or $\hat{r}_P \leq −0.2$ revealed a higher proportion of positive phenotypic and genetic correlations ($\hat{r}_P$: 8937 positive, 2593 negative) (Supplementary Figs. 1A, B and 3). This indicates that shared genetic loci affecting co-expression of transcripts are likely to have allelic effects in the same direction.

**Identification of shared eSNPs**. Estimates of $r_G$ represent the combined effects of all loci underlying the genetic control of transcripts. To provide support for our $\hat{r}_G$ estimates, we sought to identify SNPs that were associated with the expression levels of both transcripts in an $r_G$ pair. Using eQTL data from the independent LIFE-Heart study[15], we identified incidences where the top eSNP for a transcript (the SNP with the largest allelic effect) also had a significant association with the paired transcript in each $r_G$ pair. A schematic outlining our approach is given in Supplementary Fig. 4. Following quality control (see Methods), we were able to evaluate the shared effects of eSNPs for 1 198 525 $r_G$ transcript pairs. We identified 934 incidences of eSNPs associated with both transcripts at a Bonferroni significance threshold of 0.05/1198525 = $4.1 \times 10^{-8}$ (Supplementary Data 3). The mean absolute $\hat{r}_G$ of these pairs is 0.37, and thus represent pairs of transcripts that

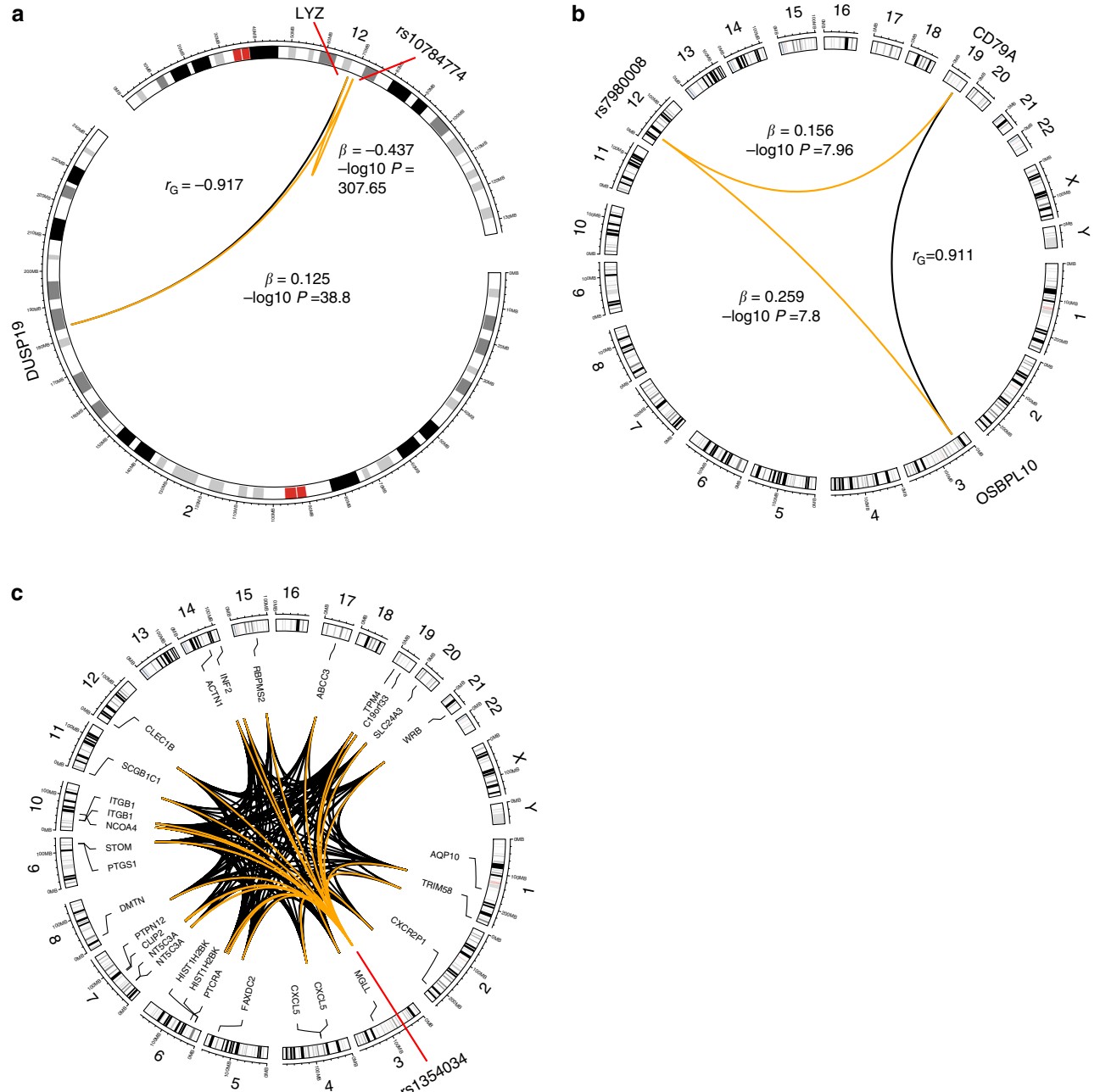

**Fig. 2** The genome-wide effects of shared eSNPs (LIFE-Heart) on genetically correlated transcript pairs. **a** (*cis/trans*) shows a transcript pair located on different chromosomes (*trans*) with a strong negative $\hat{r}_G$, where the expression of both transcripts is altered by a shared eSNP that is located on the same chromosome as one of the transcripts. *DUSP19* (chr 2) and *LYZ* (chr 12) are both regulated by a shared eSNP (rs10784774; *orange lines*), one in *cis* and one in *trans*, and have a negative $\hat{r}_G$ (−0.917; *black line*) that indicates an opposite allelic direction of effect, confirmed by the $\beta$ values. The eSNP is on the same chromosome as one transcript (*cis*). **b** (*trans/trans*) depicts the effects of a shared eSNP (rs7980008) that is located on a different chromosome to each of the two transcripts *CD79A* and *OSBPL10*, also located on separate chromosomes (19 and 3, respectively). The positive $\hat{r}_G$ between *CD79A* and *OSBPL10* (0.911) indicates their shared *trans* eSNP alters their expression in the same allelic direction, as shown by the $\beta$ values. **c** reveals the extensive transcriptional control by a single master *trans*-eSNP (rs1354034), with shared effects, that alters the expression of 30 individually genetically correlated transcripts corresponding to 26 unique genes. These 26 unique genes comprise 183 unique $\hat{r}_G$ pairs. $\beta$ is the effect size of the eSNP, with accompanying −log10-transformed *p*-value, on transcript expression from the LIFE-Heart Study data

are estimated to have a high degree of shared genetic control. Of the 934, 446 were between interchromosomal pairs, separated by an average distance of 1.6 Mb, and thus likely represent shared *cis*-eQTL. However, 488 were intrachromosomal, thereby lending further support to the hypothesis that shared genetic control of transcription exists throughout the genome. To further verify the significant shared eSNPs found in the LIFE-Heart data, we returned to our own CAGE eQTL data[20], which contained

summary statistics for 651 of the 934 eSNP-probe pairs. At a Bonferroni significant threshold of $0.05/934 = 5.4 \times 10^{-5}$, 100% replicate in CAGE (Supplementary Data 4) with 100% matched allelic direction. The result of this replication is striking and demonstrates that *trans*-eQTL that are associated with multiple transcripts are more prevalent than previously thought. It is important to note that the majority of these *trans*-eQTL would not have been identified using traditional eQTL mapping strategies

due to correction for the number of tests that are required. We further investigated the relationship between *trans*-eQTL replication and discovery effect size using summary statistics reported by Westra et al.[16]. Unsurprisingly, the large-effect *trans*-eQTL showed a higher probability of replication than those with small effects, although *trans*-eQTL with effects on multiple transcripts were consistently replicated (Supplementary Fig. 5). We examined 1975 unique SNPs that replicated in the *trans*-eQTL data and observed 645 SNPs were associated with > 1 probe (32.6%) and 58 with > 10 probes (2.9%).

To illustrate the relationship between transcript $\hat{r}_G$ pairs and their eSNPs, we investigated three examples identified by our analyses (Fig. 2). A: The $\hat{r}_G$ between a pair of transcripts tagging *LYZ* on chromosome 12 and *DUSP19* on chromosome 2 is −0.92 ($\chi^2$, $p = 3.9 \times 10^{-8}$). In the LIFE-Heart data, the top eSNP for *LYZ* is *cis*-acting rs10784774[A/G] (*t*-test, $p = 1 \times 10^{-307}$), with each copy of the A allele associated with a decrease in *LYZ* expression by 0.45 SD. We identified a shared effect of rs10784774 on *DUSP19* (*t*-test, $p = 1.6 \times 10^{-39}$), with each copy of the A allele associated in an increase in *DUSP19* expression by 0.12 SD. This *cis*/*trans* relationship between rs10784774, *LYZ* and *DUSP19* has been previously identified[21], verifying the use of genetic correlations in identifying shared genetic control between genes. B: We identified instances where the top eSNP for a transcript was in *trans* and had a significant effect on the paired transcript, also in *trans*, which we term *trans*/*trans* eSNPs. For example, *CD79A* and *OSBPL10* are located on chromosomes 19 and 3, respectively, and have an $\hat{r}_G$ of 0.91. The top eSNP in LIFE-Heart for *CD79A* is rs7980008[A/G] (*t*-test, $p = 1.26 \times 10^{-8}$), located at 30946602 bp on chromosome 12 (hg38), with each copy of the A allele associated with a 0.15 SD increase in expression of *CD79A*. rs7980008 was identified as having a significant effect on *OSBPL10* (*t*-test, $p = 1.6 \times 10^{-8}$), with each copy of the A allele increasing expression by 0.26 SD. C: SNP rs1354034[C/T], located at 56815721 bp on chromosome 3 (hg38) was identified as the top eSNP for 26 genes, 25 of which are not located on chromosome 3. At the FDR threshold of 0.05, we identify 183 significant genetic correlations between these 26 genes, with a mean $|\hat{r}_G|$ of 0.27. In all 183 pairs, rs1354034 has a significant effect (*t*-test, $p < 4.1 \times 10^{-8}$) within pairs, and 92% concordance between allelic direction and direction of the $\hat{r}_G$ (Supplementary Data 3). While rs1354034 is a known eQTL *trans*-master regulator, and is associated with blood cell phenotypes and cardiovascular diseases[22], the genetic correlation analysis applied here has enabled the identification of a large, interconnected network of genetic co-regulated genes.

Although the genetic control of most transcripts is polygenic[20], the direction of the $r_G$ estimate can be taken as an indicator of the direction of the allelic effect ($\beta$) for an eSNP associated with both transcripts; such that a positive $\hat{r}_G$ indicates the same $\beta$ direction and negative $\hat{r}_G$, the opposite $\beta$ direction. In the first instance, we asked whether the distribution of $\hat{r}_G$ and the direction of the eSNP $\beta$ for probe pairs matching (i) $+\hat{r}_G$, same sign, (ii) $+\hat{r}_G$, opposite sign, (iii) $-\hat{r}_G$, same sign, (iv) $-\hat{r}_G$, opposite sign, differed significantly from a null hypothesis of equally distributed values. Of the 934 significant shared pairs of eSNPs, we observed a significant inflation for values matching (i) 896 vs 218, and (iv) 442 vs 312, which was confirmed by a $\chi^2$ test ($\chi^2 = 298$; $p = 7.29 \times 10^{-67}$) (Supplementary Fig. 6A). We observed the same pattern if we restricted our analysis to interchromosomal $\hat{r}_G$ pairs, with (i) 447 vs 109 and (iv) 244 vs 146 ($\chi^2 = 119$; $p = 7.93 \times 10^{-41}$) (Supplementary Fig. 6B).

Finally, we investigated the relationship between eSNPs with significant effects on interchromosomal transcript pairs across the full $\hat{r}_G$ range in bins of 0.1 (Fig. 3). As expected, we observed an increase in the percentage of pairs with shared eSNPs at tails of the $r_G$ estimate distribution.

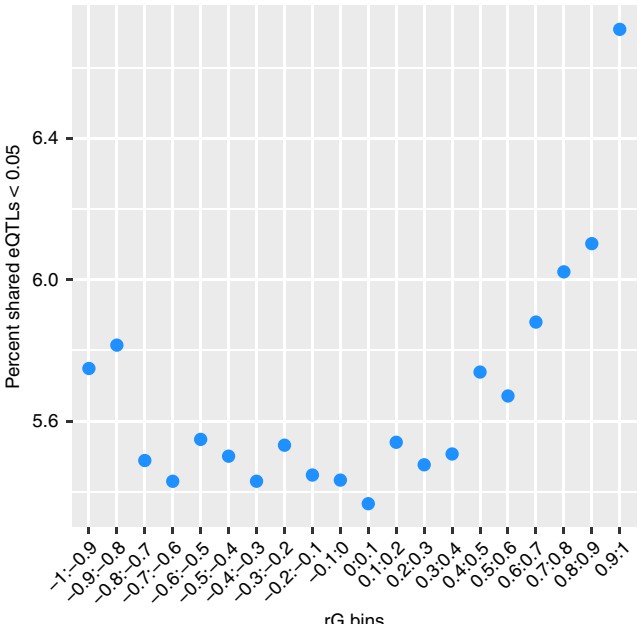

**Fig. 3** The percentage of genetically correlated *trans* transcript pairs with significant, independent, shared eSNPs in $\hat{r}_G$ bins of size 0.1. A modest enrichment of shared eQTLs, between 0.4–1.0, is evident for transcript pairs with a positive $\hat{r}_G$, with a spike between 0.9–1.0 indicating more shared eSNPs influence each other in a positive allelic direction

$\hat{r}_G$ **enrichment in interacting chromatin loci**. Our work has identified many instances of shared genetic control between pairs of transcripts located both on the same chromosomes and different chromosomes from one another. These are statistical observations based upon the identification of genetic covariance between the transcript pairs. We next sought to identify possible regulatory mechanisms that may underlie the observed genetic covariance. As transcription can be partly regulated through the spatial organisation of chromatin, permitting physical interactions between transcribed regions that are separated by large genomic distances[23, 24], we speculated that transcript pairs with significant genetic correlations may be enriched for regions that display chromatin interactions. Using Hi-C data from LCL and K562 cell lines[23, 25], we identified $\hat{r}_G$ transcript pairs that were within 0.5 Mb of interacting chromatin locations. We identified 610 pairs overlapping with intrachromosomal contacts in LCLs, and 226 intrachromosomal contacts in K562 cells. Furthermore, we identified 111 instances of interchromosomal chromatin interactions in K562 cells that overlapped with significant (FDR) $\hat{r}_G$ interchromosomal transcript pairs (Fig. 4). We observed the same results using different window sizes (1 Mb, 250 kb and 100 kb) around the chromatin interactions (Supplementary Fig. 7).

Chromatin interactions do not occur randomly throughout the genome, instead showing enrichment for certain functional genomic regions[23, 25]. Therefore, to assess empirical significance, a permutation analysis was used to verify that our observed overlap was not due to chance (Methods). We identified a significant enrichment for all window sizes (two-tailed permutation, $p < 0.001$), but a much greater enrichment for the $\hat{r}_G$ significant at the Bonferroni threshold (Supplementary Table 1). We observe the same enrichment ($p < 0.001$) if we restrict our analysis to just the 428 and 14 020 interchromosomal pairs significant at the Bonferroni and FDR threshold respectively (Supplementary Fig. 7). These results support our hypothesis that chromatin interactions could indirectly give rise to high $\hat{r}_G$ through physical co-localisation of transcripts, and thus the co-localisation of any *cis*-eQTL. Precise interactions between a locus

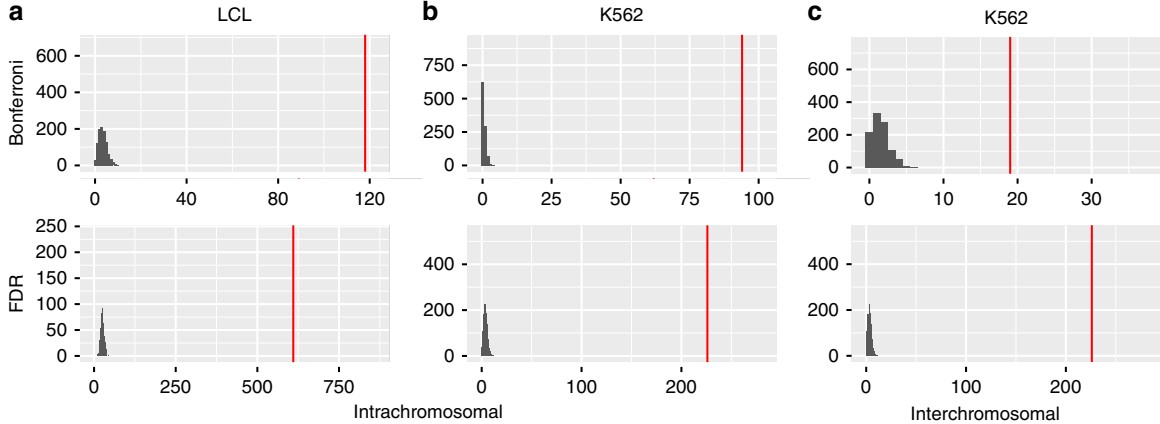

**Fig. 4** The number of overlaps between genetically correlated transcript pairs and known regions of interacting chromatin in LCLs and K562 cells. The number of transcript pairs, from the Bonferroni-corrected ($p < 1.81 \times 10^{-8}$) $\hat{r}_G$ subset ($n = 556$) or the study-wide FDR of 0.05 subset ($n = 14\,991$), that overlap known LCL and K562 chromatin interaction sites (500 kb window) are shown by *red lines*. **a** shows the intrachromosomal interactions for LCLs (Bonferroni, $n = 128$; FDR, $n = 971$), **b** shows intrachromosomal interactions for K562 cells (Bonferroni, $n = 128$; FDR, $n = 971$), and **c** shows interchromosomal interactions for K562 cells (Bonferroni, $n = 428$; FDR, $n = 14\,020$). The null distribution represented as a *grey histogram* was obtained by randomly sampling each dataset 1000 times for each window size

| **Table 2 Proportion of observed transcript pairs at standard deviations (σ) defined under null hypothesis (H$_0$: $\hat{r}_G = 0$)** | | | | | |
|---|---|---|---|---|---|
| σ | $\hat{r}_G$ | **%−$\hat{r}_G$** | | **% + $\hat{r}_G$** | |
| | | **Exp** | **Obs** | **Exp** | **Obs** |
| 2 | 0.48 | 2.2 (60 620) | 10.42 (287 199) | 2.2 (60 620) | 10.55 (290 783) |
| 3 | 0.72 | 0.1 (2755) | 3.72 (102 635) | 0.1 (2755) | 3.78 (104 250) |
| 4 | 0.96 | 0.01 (275) | 1.44 (39 651) | 0.01 (275) | 1.47 (40 585) |

Each σ value corresponds to the adjacent |$\hat{r}_G$| threshold. Actual expected and observed number of transcript pairs are shown in parentheses.

harbouring a specific regulatory eSNP and another genomic locus may be detectable using a very small interaction window of 1 kb. We repeated the analysis using a window of 1 kb around an interaction locus for intrachromosomal contacts in LCL and K562 cells, and interchromosomal contacts in K562 cells, for significant (FDR) $\hat{r}_G$ pairs. However, for all three analyses, we did not detect any interactions with a 1 kb window size, and 1000 matched permutations for each analysis also returned zero interactions. Tables of transcript pairs located in LCL and K562 chromatin interaction locations are available in the supplementary data as well as online at http://computationalgenomics.com.au/shiny/rg/.

**Networks of genetically correlated transcripts**. Biological networks can reveal interesting and potentially unexpected relationships between interacting genes. Genetic regulation often occurs at a network level, and genetic regulatory networks (GRNs) are useful for identifying master regulator genes that alter the expression of a set of molecular targets, and highlighting functional pathways. We hypothesised that genetically correlated transcripts would form the basis of GRNs and reveal the connectivity (strength and allelic direction) between themselves and their $r_G$ pairs, as well as identify novel relationships between genes.

Under a null model of H$_0$: $r_G = 0$, we estimated the empirical $r_G$ coefficient threshold corresponding to the 2nd, 3rd and 4th standard deviation (σ) from the mean. To identify whether subsets of $\hat{r}_G$ estimates were associated with disproportionately high shared levels of genetic control, we calculated the proportion of probe pairs at 2, 3 and 4σ from the null mean (|$\hat{r}_G$| ≥ 0.48, 0.72

and 0.96). We observed a 4-fold, 24-fold and 36-fold enrichment for the 2nd, 3rd and 4th SDs, respectively, compared to the number of correlated transcript pairs expected by chance (Table 2, Supplementary Table 2).

To understand the connectivity of genetically correlated transcripts, we constructed network graphs based on transcripts where we observed an enrichment in high $\hat{r}_G$ pairs with other transcripts. Since multiple transcripts exist for many genes, we first determined which transcripts were the most connected to other transcripts, indicated by the number of pairings with high |$\hat{r}_G$|. For each transcript, we observed that the majority of connections (94.93%) were interchromosomal, with the remaining 5.07% of connections intrachromosomal. To isolate transcripts with strong biological and functional importance, we selected those with the highest number of connections above a threshold of |≥ 0.72|, which represents ±3σ from the mean of 0 under a null model where H$_0$ : $r_G = 0$. Figure 5a shows connectivity details for the top 50 connected transcripts ±3σ. Using the most connected gene (ZNF536) as an example, Fig. 5b reveals the extent of the genome-wide distribution of transcripts that are genetically correlated with ZNF536, while Fig. 5c shows $r_G$ transcripts located only on the same chromosome as ZNF536. Furthermore, we investigated the functional enrichment of each of the top 50 connected transcripts using a pathway analysis approach. Our results reveal significant (Fisher's exact test, $p < 0.01$) functional enrichment for 50 identified networks (Supplementary Table 3). For example, the pathway analysis for transcript $r_G$ pairs arising from ZNF536 revealed an enrichment of immune and inflammatory response processes including

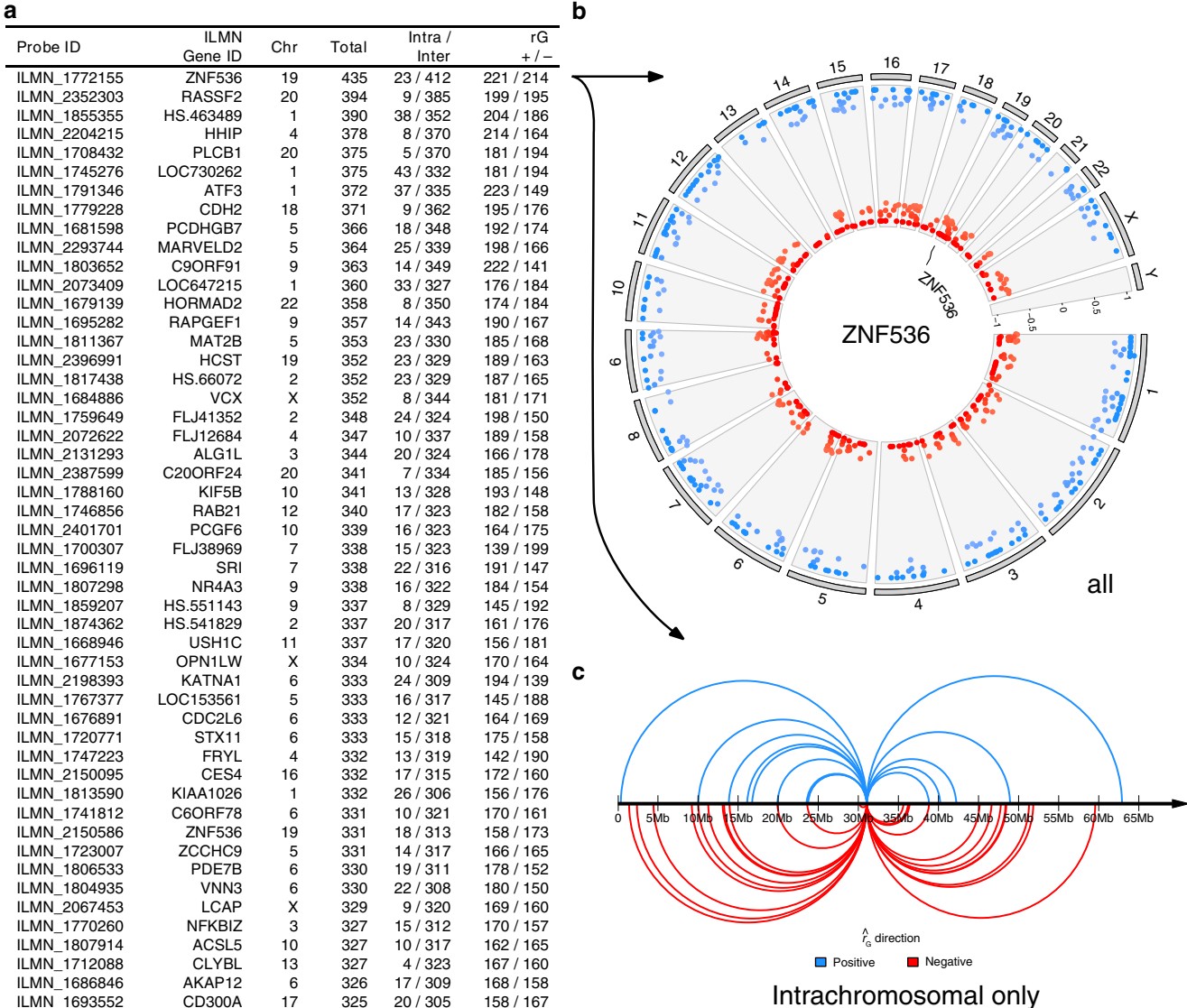

| Probe ID | ILMN Gene ID | Chr | Total | Intra / Inter | rG + / − |
|---|---|---|---|---|---|
| ILMN_1772155 | ZNF536 | 19 | 435 | 23 / 412 | 221 / 214 |
| ILMN_2352303 | RASSF2 | 20 | 394 | 9 / 385 | 199 / 195 |
| ILMN_1855355 | HS.463489 | 1 | 390 | 38 / 352 | 204 / 186 |
| ILMN_2204215 | HHIP | 4 | 378 | 8 / 370 | 214 / 164 |
| ILMN_1708432 | PLCB1 | 20 | 375 | 5 / 370 | 181 / 194 |
| ILMN_1745276 | LOC730262 | 1 | 375 | 43 / 332 | 181 / 194 |
| ILMN_1791346 | ATF3 | 1 | 372 | 37 / 335 | 223 / 149 |
| ILMN_1779228 | CDH2 | 18 | 371 | 9 / 362 | 195 / 176 |
| ILMN_1681598 | PCDHGB7 | 5 | 366 | 18 / 348 | 192 / 174 |
| ILMN_2293744 | MARVELD2 | 5 | 364 | 25 / 339 | 198 / 166 |
| ILMN_1803652 | C9ORF91 | 9 | 363 | 14 / 349 | 222 / 141 |
| ILMN_2073409 | LOC647215 | 1 | 360 | 33 / 327 | 176 / 184 |
| ILMN_1679139 | HORMAD2 | 22 | 358 | 8 / 350 | 174 / 184 |
| ILMN_1695282 | RAPGEF1 | 9 | 357 | 14 / 343 | 190 / 167 |
| ILMN_1811367 | MAT2B | 5 | 353 | 23 / 330 | 185 / 168 |
| ILMN_2396991 | HCST | 19 | 352 | 23 / 329 | 189 / 163 |
| ILMN_1817438 | HS.66072 | 2 | 352 | 23 / 329 | 187 / 165 |
| ILMN_1684886 | VCX | X | 352 | 8 / 344 | 181 / 171 |
| ILMN_1759649 | FLJ41352 | 2 | 348 | 24 / 324 | 198 / 150 |
| ILMN_2072622 | FLJ12684 | 4 | 347 | 10 / 337 | 189 / 158 |
| ILMN_2131293 | ALG1L | 3 | 344 | 20 / 324 | 166 / 178 |
| ILMN_2387599 | C20ORF24 | 20 | 341 | 7 / 334 | 185 / 156 |
| ILMN_1788160 | KIF5B | 10 | 341 | 13 / 328 | 193 / 148 |
| ILMN_1746856 | RAB21 | 12 | 340 | 17 / 323 | 182 / 158 |
| ILMN_2401701 | PCGF6 | 10 | 339 | 16 / 323 | 164 / 175 |
| ILMN_1700307 | FLJ38969 | 7 | 338 | 15 / 323 | 139 / 199 |
| ILMN_1696119 | SRI | 7 | 338 | 22 / 316 | 191 / 147 |
| ILMN_1807298 | NR4A3 | 9 | 338 | 16 / 322 | 184 / 154 |
| ILMN_1859207 | HS.551143 | 9 | 337 | 8 / 329 | 145 / 192 |
| ILMN_1874362 | HS.541829 | 2 | 337 | 20 / 317 | 161 / 176 |
| ILMN_1668946 | USH1C | 11 | 337 | 17 / 320 | 156 / 181 |
| ILMN_1677153 | OPN1LW | X | 334 | 10 / 324 | 170 / 164 |
| ILMN_2198393 | KATNA1 | 6 | 333 | 24 / 309 | 194 / 139 |
| ILMN_1767377 | LOC153561 | 5 | 333 | 16 / 317 | 145 / 188 |
| ILMN_1676891 | CDC2L6 | 6 | 333 | 12 / 321 | 164 / 169 |
| ILMN_1720771 | STX11 | 6 | 333 | 15 / 318 | 175 / 158 |
| ILMN_1747223 | FRYL | 4 | 332 | 13 / 319 | 142 / 190 |
| ILMN_2150095 | CES4 | 16 | 332 | 17 / 315 | 172 / 160 |
| ILMN_1813590 | KIAA1026 | 1 | 332 | 26 / 306 | 156 / 176 |
| ILMN_1741812 | C6ORF78 | 6 | 331 | 10 / 321 | 170 / 161 |
| ILMN_2150586 | ZNF536 | 19 | 331 | 18 / 313 | 158 / 173 |
| ILMN_1723007 | ZCCHC9 | 5 | 331 | 14 / 317 | 166 / 165 |
| ILMN_1806533 | PDE7B | 6 | 330 | 19 / 311 | 178 / 152 |
| ILMN_1804935 | VNN3 | 6 | 330 | 22 / 308 | 180 / 150 |
| ILMN_2067453 | LCAP | X | 329 | 9 / 320 | 169 / 160 |
| ILMN_1770260 | NFKBIZ | 3 | 327 | 15 / 312 | 170 / 157 |
| ILMN_1807914 | ACSL5 | 10 | 327 | 10 / 317 | 162 / 165 |
| ILMN_1712088 | CLYBL | 13 | 327 | 4 / 323 | 167 / 160 |
| ILMN_1686846 | AKAP12 | 6 | 326 | 17 / 309 | 168 / 158 |
| ILMN_1693552 | CD300A | 17 | 325 | 20 / 305 | 158 / 167 |

**Fig. 5** Visualising highly connected, genetically correlated transcripts. **a** shows a table of transcripts with the highest number of connections above the threshold of $3\sigma$ ($|\hat{r}_G| \geq 0.72$), their chromosome number, total number of connections above the threshold, proportion of intrachromosomal/interchromosomal loci, and proportion of $\hat{r}_G$ with directionality. Using the example of *ZNF536*, the gene with the highest number of connections at $3\sigma$, the transcriptome-wide connections at $\geq 3\sigma$ are visualised as a *circle plot*. The *circle plot* shows the genomic position and associated $\hat{r}_G$ values for each transcript that is correlated with *ZNF536* at $\geq 3\sigma$, represented as a heatmap ranging from −1 to 1 (*red* to *blue*) (**b**). **c** shows intrachromosomal connections between transcript pairs at $\geq 3\sigma$. Positive and negative $r_G$ estimates are shown in *blue* and *red*, respectively. The complete table of transcript connections $\geq 3\sigma$ is available in the supplementary materials. Circle (**b**) and arc (**c**) plots can be generated at the desired $\hat{r}_G$ threshold for genes in our data set via the browser interface

positive regulation of NLRP3 inflammasome complex assembly ($p = 0.002$), regulation of immune response ($p = 0.002$), positive regulation of interferon-alpha production ($p = 0.002$), B cell proliferation involved in immune response ($p = 0.003$) and interleukin-1 beta secretion ($p = 0.003$).

By graphing interactions between transcript pairs, we were able to assess differences in genetic control for alternative isoform networks. In cases such as *ZNF536*, the connections for each transcript isoform, both independent and shared, can be distinguished. The resulting graph network depicted in Supplementary Fig. 8 shows two distinct clusters of $\hat{r}_G$ connectivity formed around the two *ZNF536* transcripts (tagged by probes ILMN 1772155 and ILMN 2150586 on the Illumina HT12 array). Notably, each transcript has an associated set of connected transcripts that are predominantly distinct from one another. However, there are several transcripts whose regulation of expression appears to be shared by both *ZNF536*

transcripts. For example, the $r_G$ sign direction for *RAB21* was in opposite directions for the two *ZNF536* isoforms. This suggests that the loci that have shared effects on *RAB21* and *ZNF536* have on average opposite allelic effects for the alternative *ZNF536* isoforms. We have produced a web application to visualise and download networks for each gene or transcript (http://computationalgenomics.com.au/shiny/rg/).

Because of the complex and highly connected nature of the transcriptome, we expected a large number of genetically correlated transcript pairs to exhibit many connections. To establish the baseline connectivity of genetically correlated transcripts, and to identify those with a significantly greater number of connections than expected by chance, we calculated the expected number of transcript connections at two and three $\sigma$ and compared them to our observed data. Under the null hypothesis ($H_0: r_G = 0$), the expected number of $\hat{r}_G > \pm 2$ and $3\sigma$ per transcript is 113 and 7, respectively. In our data, we observed

2317/2468 unique transcripts ($|\hat{r}_G| \geq 0.48$) with > 113 connections (mean = 468), and 2397/2445 ($|\hat{r}_G| \geq 0.72$) with > 7 connections (mean = 169) (Supplementary Fig. 9). For unique transcripts above $4\sigma$ ($|\hat{r}_G| \geq 0.96$), the expected number of connections is < 1. The high proportion of strongly genetically correlated transcripts with a much greater number of $r_G$ pairs than expected suggests the GRNs we have identified using estimates of $r_G$ have strong, biologically important relationships.

**High $\hat{r}_G$ pairs are enriched for transcription factors.** Genes that exhibit a large number of significant genetic correlations with other genes (hub genes) may perform an important role in the regulation of their paired genes, with one possibility being hub genes encode transcription factors. An enrichment of transcription factors in $\hat{r}_G$ pairs could reveal novel binding partners and simultaneously provide useful information about the strength, direction, and possibly the origin of their shared genetic regulation. To address this, we queried whether there was an enrichment of transcription factors and their predicted binding sites amongst the genes above null thresholds of 2 and $3\sigma$ of $\hat{r}_G$. We tested unique genes at each threshold against a list of unique human TFs constructed from the FANTOM 4 data[26] and Vaquerizas et al.[27]. From 2671 unique TFs, 234 are expressed in our data set of 2330 unique genes. For the unique genes above the $2\sigma$ (i.e. with > 113 $|r_G| >0.48$), we found 229/234 TFs in 2193 unique gene IDs (Hypergeometric test, $p = 1.14 \times 10^{-3}$), and for unique genes above $3\sigma$ with > 7 $|r_G|$ greater than 0.72, we found 231/234 TFs in 2266 unique gene IDs (Hypergeometric test, $p = 3.62 \times 10^{-2}$), revealing a strong enrichment for transcription factor-related connections between genetically correlated probes whose $|\hat{r}_G| \geq 0.48$ or 0.72, respectively. The list of enriched TFs is available in Supplementary Table 4.

## Discussion

To better understand the relationship between the co-expression and genetic co-regulation of transcripts, we examined the phenotypic and genetic correlations between expressed and highly heritable transcripts in whole blood. We used the genotype and expression data of 1748 unrelated individuals of European ancestry to determine the average effect of all common loci on shared genetic mechanisms controlling gene expression. Our results show that the mean levels of shared genetic control between transcripts is close to zero, although there were a large number of significant transcript pairs with both positive and negative $\hat{r}_G$. We show that the proportion of interchromosomal transcript pairs is greater than intrachromosomal pairs, implying that the primary mode of shared genetic regulation is through distal mechanisms. Transcripts with the highest number of paired (connected) transcripts are likely to represent a core regulatory mechanism. We examined the most highly connected transcripts above $|\hat{r}_G| \geq 0.96$ (empirical $4\sigma$) to visualise their connectivity and potential to form discrete transcriptional clusters. These transcript pairs were enriched for highly connected transcription factors, reflecting their widespread regulation of the expression of large numbers of genes. We traced the regulatory connectivity of ZNF536, a ubiquitously expressed member of the Krüppel C2H2 zinc finger transcription factor family[28]. ZNF536 is expressed in whole blood as two isoforms available on the Illumina HT12 array, both of which are highly correlated with many transcripts. We constructed a network graph for ZNF536 and demonstrated that each transcript isoform is genetically correlated with discrete groups of transcripts. Where both isoforms were correlated with the same transcript, such as RAB21, we observed opposite sign directions of $\hat{r}_G$, implying that individual ZNF536 isoforms exert different regulatory effects on certain targets. In general, little is known about C2H2 zinc finger transcription factors and their regulatory partners[29], including ZNF536; however, our analyses demonstrate that using genetic correlations to uncover novel transcriptional networks can greatly improve our understanding of the strength and direction of genetically co-regulated transcripts.

Because mixed model methods such as bivariate GREML[3] estimate $r_G$ by capturing the effect of all common SNPs rather than each SNP individually, we can only infer the averaged effects of SNP-mediated regulation on a given pair of transcripts. Thus, to provide further verification of our $r_G$ estimates, we linked transcript pairs with high $\hat{r}_G$ with significant shared eSNPs from the independent LIFE-Heart eQTL study[15]. By demonstrating the overlap of known peripheral blood mononuclear cell (PBMC) eQTLs with pairs of transcripts identified using an orthogonal method to capture the total shared genetic effects, we are able to identify many novel instances of trans-eQTL that would otherwise have been missed by traditional eQTL mapping strategies. These results support the evidence that the majority of genetic variance for gene expression is located on trans chromosomes[14, 19]. Future work aiming to determine the effect of a given SNP on transcript expression would benefit from considering the use of nuclear run-on techniques such as GRO-Seq[30] or NET-Seq[31], which, coupled with chromatin immunoprecipitation data, provide higher resolution of transcriptional activity over time.

By examining the correlation structure of expressed transcripts, both novel and established genetic co-regulatory events can be identified in an agnostic manner. This approach can be extended so that the exact nature of the complex regulatory interactions between many highly correlated transcripts, such as long-distance chromatin interactions or trans-eQTL, can be confidently determined. In the scenario presented in Fig. 2a, a single cis-eSNP alters the expression of a proximally located transcript (LYZ) and also drives the expression of another gene, DUSP19, in trans, termed a cis/trans shared eSNP. Our data uncovered the genetic co-regulatory relationship between LYZ and DUSP19 as a highly negative estimate of $r_G$, indicating opposite directions of allelic effect by a genetic variant. Using genetic correlations alone, we observed strong co-regulation of expression between LYZ and DUSP19, the exact nature of which was even more precisely determined when combined with independent eQTL and Hi-C data. By overlapping our findings with eQTL data, we identified the eSNP (rs10784774) with the strongest association for at least one of the two transcripts, with SNP allelic effects consistent with the $\hat{r}_G$ sign direction. We also discovered instances in which the eSNP with the strongest association to a given transcript was a trans/trans eSNP (Fig. 2b). From the perspective of genetic variants that alter events such as transcription factor binding, the ability to use an unbiased approach to detect which downstream gene targets are most likely to be affected is very powerful, enabling the genome-wide dissection of complex, co-regulated transcriptional networks (Fig. 2c). Our study shares an approach similar to multivariate eQTL analyses[12, 32, 33], where an eSNP is shown to control the expression of multiple transcripts, and the combination of both methods would greatly increase the likelihood of identifying such cases.

Our data also revealed a significant overlap with known regions of interacting chromatin. This suggests the possibility that some strongly correlated pairs of expressed transcripts undergo shared genetic regulation resulting from spatial organisation or remodelling of chromatin that brings these transcripts into close proximity. We observed transcript pairs that overlapped with eQTL and Hi-C data, as well as being enriched for transcription factors. That many of the observed chromatin interactions overlap with $\hat{r}_G$ transcript pairs in regions less than 1 Mb suggests that those transcript pairs are likely regulated by shared eQTL; however, it is difficult to disentangle the precise mechanisms of

regulation. This supports the notion that the regulatory relationship between transcripts is not limited to a single mechanism, but is multi-faceted and likely temporospatially dynamic.

In this study, we estimated the amount of shared genetic control of expressed transcripts within a single tissue to gain insight into the mechanisms that control gene expression in a tissue-specific context. By estimating the genetic correlation between pairs of expressed transcripts, the genetic contribution to co-expression of genes can be ascertained, which allows the determination of the size, density and regulatory direction in genetic regulatory networks built around a gene or transcript of interest. We have shown that a comprehensive analysis of expressed transcripts using bivariate GREML can reveal which genetic mechanisms of transcriptional regulation underlie expression patterns, and promote the discovery of novel regulatory links between transcripts. The nature of the regulatory mechanism underlying each correlated transcript pair can be further investigated when overlaid with functional genomics studies such as Hi-C, chromatin interaction and eQTL data for the relevant cell or tissue types.

## Methods

**Cohort data.** Gene expression levels, measured from whole blood, and genotype data were available in 1748 unrelated individuals of European descent. These samples are a subset from the Consortium for the Architecture of Gene Expression (CAGE) cohort, which includes both multi-ethnic and related individuals. Full details of CAGE are given in Lloyd-Jones et al[20], but briefly, the CAGE data set includes gene expression data for 38 624 expression probes measured on Illumina HT12-v4.0 BeadChip arrays, along with genotypes for 8 242 192 SNPs with minor allele frequency (MAF) 0.01 imputed to the 1000 Genome (phase 1v3) reference panel. Gene expression data has been normalised and corrected for known and hidden batch effects, and blood cell type composition. Transcripts tagged by the Illumina probes were annotated using custom mappings from the illuminaHumanv4.db package. We selected unrelated individuals from CAGE based on an identity-by-state genetic relatedness threshold of 0.05, and European ancestry through the projection of the first two principal components from CAGE genotypes against HapMap 3 ancestry cohorts. As the ability to accurately estimate $r_G$ is partly a function of the heritability of the transcripts, we restricted our analysis to those probes with $\hat{h}_g^2 > 0.25$, where $\hat{h}_g^2$ is an estimate of the proportion of phenotypic variance that can be attributed to the common genetic variance captured from imputed genotype data. Although the stringent $\hat{h}_g^2$ threshold substantially reduced the number of transcripts analyzed, this approach ensured that there was a high-degree of genetic variance that could potentially be shared between pairs of transcripts, and minimised the average standard error (SE) of $\hat{r}_G$ across the study (Supplementary Figs. 10 and 11). Finally, we retained only those transcripts that mapped to an annotated RefSeq gene. After filtering, the final data taken forward comprised gene expression levels for 2469 highly heritable transcripts, mapping to 2330 unique genes, each of which were measured in 1748 unrelated individuals.

**Estimating genetic correlations.** We estimated the genetic correlation ($r_G$) for each pairwise combination of the 2469 transcripts using a mixed-model bivariate GREML algorithm. GREML is described in detail in Lee et al.[3], but briefly, for a pair of transcripts $i$ and $j$, the linear mixed-effects model can be written as, $y_i = g_i + e_i$ and $y_j = g_j + e_j$, where $y_i$ and $y_j$ are $n*1$ vectors of normalised gene expression levels for transcripts $i$ and $j$ respectively; $g_i$ and $g_j$ are $n*1$ vectors of random polygenic effects with $g_i \sim N(0, \sigma_{g_i}^2 \mathbf{A})$ and $g_j \sim N(0, \sigma_{g_j}^2 \mathbf{A})$, with $\mathbf{A}$ the genetic relationship matrix (GRM) estimated from 8 242 192 SNPs; and $e_i$ and $e_j$ $n x 1$ vectors of residuals with $e_i \sim N(0, \sigma_{e_i}^2 \mathbf{I})$ and $e_j \sim N(0, \sigma_{e_j}^2 \mathbf{I})$, with $\mathbf{I}$ as the incidence matrix. The variance–covariance matrix for $y_i$, $y_j$ is defined as,

$$V = \begin{bmatrix} \mathbf{A}\sigma_{g_i}^2 + \mathbf{I}\sigma_{e_i}^2 & \mathbf{A}\sigma_{g_i,g_j} \\ \mathbf{A}\sigma_{g_i,g_j} & \mathbf{A}\sigma_{g_j}^2 + \mathbf{I}\sigma_{e_j}^2 \end{bmatrix}$$

with $\sigma_{g_i}^2$, $\sigma_{g_j}^2$ and $\sigma_{g_i,g_j}$ the genetic variance and covariance for transcripts $i$ and $j$, respectively. The genetic variance and covariance components were estimated using GREML, implemented in GCTA software, with $\hat{r}_G$ constrained to the boundaries of $-1$ and $+1$. The genetic correlation between the pair of transcripts was estimated as,

$$\hat{r}_G = \frac{\hat{\sigma}_{g_i,g_j}}{\sqrt{\hat{\sigma}_{g_i}^2 \hat{\sigma}_{g_j}^2}}$$

We calculated a test statistic by dividing the square of the $\hat{r}_G$ by its approximate sampling variance and calculated a p-value assuming a $\chi^2$ distribution with one degree of freedom. The SE of $\hat{\sigma}_{g_i,g_j}$, $\hat{\sigma}_{g_i}^2$ and $\hat{\sigma}_{g_j}^2$ are estimated directly, and the SE of $\hat{r}_G$ calculated through a Taylor series approximation.

**Pathway analysis of highly connected networks.** Pathway analysis was performed on each of the $\hat{r}_G$ networks formed by the 50 most highly-connected expression transcripts, listed in Fig. 5a. For each transcript, the connected $\hat{r}_G$ pairs $\geq 0.72$ (3 SD from the mean) were analyzed with TopGO, using the 'weight01' parameter to account for GO term hierarchy and a background list of 2043 unique genes in the 3 SD $\hat{r}_G$ data. The gene enrichments for each GO term were calculated by a Fisher's exact test with a threshold of $p < 0.01$.

**LIFE-Heart eQTL.** Using eQTL data generated from PBMC gene expression in an independent cohort, we investigated the relationship between shared effects of SNPs on both transcripts in $\hat{r}_G$ pair, and the magnitude of $\hat{r}_G$. The independent eQTL results consisted of summary statistics from an additive allelic model regressing SNP genotypes against normalised expression levels, using data from 2104 unrelated individuals of European descent from the LIFE-Heart study. Expression levels were measured on Illumina HT12-v4.0 arrays, and genotypes imputed to the 1000 Genome (phase 1v3) reference panel. Expression levels were normalised and corrected for known and hidden batch effects, and blood cell type composition. Using this resource, we performed the following analyses: For each pair of transcripts ($i$,$j$) used to estimate $\hat{r}_G$, we identified the SNP with the strongest association (smallest p-value below a threshold of $1 \times 10^{-6}$) for each transcript. These SNPs were termed eSNP$_i$ and eSNP$_j$ respectively. We did not restrict our analysis to eSNPs within a certain distance from the transcript, meaning that the eSNP for a given transcript could potentially be located on a different chromosome. We then extracted the summary statistics for the association between eSNP$_i$ and transcript $j$ and eSNP$_j$ and transcript $i$. This study design is summarised in Supplementary Fig. 4. Additionally, we required the following criteria to be met in the LIFE-Heart data: transcript had to be expressed in at least 5% of the samples, and not associated with technical batches.

**Network graph analysis.** Network graphs are a powerful way to visualise the connectivity between genes or transcripts, and are useful for identifying novel connections and clusters of functionally related mRNAs. We hypothesised that transcripts that displayed strong genetic correlations with large numbers of other transcripts would form discrete clusters, or networks, illustrating the regulatory direction of effect on expression at the isoform level. Since, for a given transcript, we have estimated the $r_G$ with 2468 paired transcripts, under a null where ($H_0 : r_G = 0$), we expect to observe a fraction of pairs with high $r_G$ due to sampling variance. Thus, to identify transcripts that are the hub in a network displaying high $\hat{r}_G$ with transcripts across the genome, we first calculated the expected sampling variance under $H_0 : r_G = 0$, using an approximation derived by, $var(\hat{r}_G | r_G = 0) \approx \frac{1}{\overline{h}_{g_i}^2 \overline{h}_{g_j}^2 N^2 var(\mathbf{A}_{ij})} \approx 0.058$. Where $\overline{h}_{g_i}^2$ and $\overline{h}_{g_j}^2$ are the mean heritabilities for transcripts $i = j = 0.375$, $N$ is the sample size $=1748$, and $var(\mathbf{A}_{ij})$ is the variance of the off-diagonals of the GRM $=4.0 \times 10^{-5}$. Assuming a Gaussian distribution, with mean of 0, under a null hypothesis, we are able to calculate the expected number of $\hat{r}_G$ observations $\pm 3\sigma$ from the mean as $m \times (1 - 0.9973)$, where $m$ is the number of $\hat{r}_G$ estimates. Therefore, for a given transcript, under $H_0$: $r_G = 0$, we would expect to observe 6.6 $\hat{r}_G > 0.72 < -0.72$. For each transcript, we determined the number of $\hat{r}_G$ that were observed to be $\pm 3\sigma$ from 0. We reasoned that a transcript exhibiting a large number of high $\hat{r}_G$ connections represents a co-regulation hub, whereby the genetic variants influencing their expression levels are shared with large number of other transcripts. To visualise these transcript-level networks, we graphed connections between each probe using the Fruchterman-Reingold algorithm in the *igraph* package for R. Transcripts were annotated using Illumina probe IDs.

**Transcription factor enrichment.** Transcription factors are major drivers of gene expression and have been identified as the mechanisms responsible for observable gene co-expression networks. This led us to test the hypothesis that genes tagged by transcripts that form pairs with high $\hat{r}_G$ would be enriched for known transcription factors. Unique transcription factors from the FANTOM 4 data set and Vaquerizas et al. were combined into a single reference set of 2671 TF genes. A hypergeometric test was used to determine whether there was a significant enrichment of transcription factors in our data for transcript pairs with a high $|\hat{r}_G|$: 0.48 ($2\sigma$) or 0.72 ($3\sigma$). A background list of 2330 unique genes, and 234 expressed and heritable TFs present in our data set were used in the analysis.

**Chromatin interactions.** In this study, we refer to the co-localisation of two chromosome regions as a chromosome interaction. We cross-referenced our transcript positional locations from the $\hat{r}_G$ pairs with chromosome interactions using (i) the Bonferroni-corrected set of 556 transcript pairs (128 intrachromosomal, 428 interchromosomal), and (ii) 14 991 transcript pairs below a study-wide FDR of 0.05 (971 intrachromosomal, 14 020 interchromosomal) in windows of 100, 250, 500 kb and 1 Mb. We did not include pairs where transcripts were located at distances smaller than the queried window to remove *cis*-effects. LCL data generated by Lieberman-Aiden et al.[23] was obtained from NCBI GEO (GSE18199) and K562 chromatin contact data was obtained from the study of Lan et al.[23]. Hi-C QC and filtering was performed using the Homer Hi-C[34] analysis software with the default parameters. Based on methods detailed by Lan et al., the threshold of self-

ligation events in the LCL data was set to 20 kb, and those interaction loci within < 20 kb of each other were removed. We performed simulations to test if our observed overlaps were greater than those expected by chance. We designed the simulations to ensure that the null distribution was of chromosome interactions between loci that matched the genomic regions from which the transcripts were located. To do this we randomly shuffled transcript $i$ with respect to transcript $j$ for the list of transcript pairs, creating permuted pairs for each of 128, 971 and 14 020 genomic positions. We then tested for overlap using windows of 100 kb, 250, 500 kb and 1 Mb between the permuted list and the map of chromosome interactions and then repeated the random shuffling process. We performed 1000 such permutations to generate a null distribution of chromosome interaction overlaps.

**Data availability**. The genetic correlations data that support the findings of this study are available from http://computationalgenomics.com.au/shiny/rg/. The genotype and gene expression data can be made available to researchers through the CAGE consortium. Contact J.E.P. for details.

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

## Acknowledgements
This work was supported by the Australian National Health and Medical Research Council (NHMRC) grants (1046880, 1083405, 1107599, 1083656, 1078037, 1078399) and the Sylvia and Charles Viertel Charitable Foundation. LIFE-Heart is a part of the LIFE Research Center for Civilization Diseases. LIFE is funded by means of the European Union, by the European Regional Development Fund (ERFD), the European Social Fund and by means of the Free State of Saxony within the framework of the excellence initiative.

## Author contributions
S.W.L. performed the analysis and wrote the paper, J.E.P. designed the study, contributed to method development and wrote the paper. L.R.L.-J., A.H., H.K., G.H., J.Y., K.S., J.Z., A.M., E.T.D., G.G., T.D.S., J.T., M.S., G.W.M., T.E. and P.M.V. contributed data. H.K. contributed to data analysis. L.R.L.-J., A.H., J.Y. and P.M.V. contributed to method development.

## Additional information

**Competing interests:** The authors declare no competing financial interests.

