## [Peer Review File · Nature Communications]

Reviewers' comments:

Reviewer #2 (Remarks to the Author):

The manuscript by Powell and colleagues studied the interesting problem of genetic correlation (shared genetic basis) of gene expression traits. Comparing with the typical eQTL analysis, this is a very novel idea and can potentially provide more information of the underlying gene regulatory networks, not available from the eQTL perspective. The authors were able to identify tens of thousands of genetically correlate transcript pairs, and characterized the nature and pattern of such correlation, e.g. many of the cis-pairs (close in chromosome locations) correspond to chromatin interactions identified in Hi-C. The paper should be of broad interest to readers studying eQTL and gene regulation.

Major comments:

Replication of genetic correlations in an independent data: The authors apparently have two datasets, their own eQTL data (CAGE?) and LIFE-Heart study. A natural question is whether they can replicate their r_G obtained in CAGE, in the second dataset. The authors seem to do something along the line of replication, when they described their study of shared eSNPs in the section "Identification of shared eSNPs" of Results, but a more direct way is just to compare r_G from two studies.

Identification of shared eSNPs: in this section of Results, there are several issues about the analysis in the first paragraph. First, Figure S3 is a bit strange. The labels in S3A do not match the text, e.g. there is an arrow from eSNP_i to eSNP_{j,i}, but based on the text, it seems that it should be eSNP_i to eSNP_{i,j}. Second, there is no explanation of Figure S3B and C in the text. Why did the authors do this? Lastly, a natural question to ask here is that, given eSNP_i (the strongest eSNP of probe i), how often it is also associated, at some appropriate threshold, with a genetically correlated transcript? How does this proportion compare with random expectation?

In the end of the first paragraph of page 5, the authors reported 100% replication. As I understood, among 934 eSNPs, about half represent trans-eQTL. This result is very striking. For all the trans-eQTL studies in literature, the reviewer's impression is that often trans-eQTL are tissue-specific and have low rates of replication. Presumably the 100% replication of trans-eQTL is to some extent due to the fact that these eQTL are associated with at least a pair of probes, thus more likely to be real eQTL. Assuming this is indeed the case, the authors should somehow highlight the results and perhaps investigate a bit more to address a more general point/hypothesis: that large effect trans-eQTL, associated with multiple transcripts, can often be replicated.

The section "Genetically correlated transcripts are enriched in regions of interacting chromatin": there seems a clear enrichment of probe pairs that correspond to interacting pairs in Hi-C. However, the reviewer wonders whether this is really due to chromatin interactions. One expects that the regulatory loci (eQTL) interact with their targets, but we do not necessarily expect that two transcripts form chromatin interactions with each other. So the results seem somewhat puzzling here. Could the enrichment here be simply due to that the correlated pairs are often close to each other? Have the authors properly control for distance when obtaining the null distribution?

Network connectivity analysis at the end of page 7: is the analysis based on probes or transcripts? If it is probe-based, then the estimated connectivity may be inflated, as a single correlated transcript/gene pair may generate many probe pairs. Has this been properly addressed? The authors said that 2,317/2,468 probes have more than 113 connections at $r_G > 0.48$. What does the result say here - what's the biological significance? What do we expect by chance?

The section "Highly connected transcripts are enriched for transcription factors": the description is

not clear. If I understand correctly, the authors found that 229/234 TFs have more than 113 connections. To obtain the p-value of $1.14E-3$, what was this compared against, 2317/2468?

In Table 2: h^2 seem really high, for some probes, h^2 is close to 1. Most estimates of expression heritability seem to be on the lower end, e.g. see Wright et al, Nature Genetics, 2014. Why is this the case? If the authors estimate h^2 from their genes, does the distribution agree with the distribution reported in other papers?

It would be helpful if the authors can do more analysis and validation on the predicted transcript pairs. One obvious question is that are these pairs tend to be more functionally related than expected by chance (perhaps better, expected based on phenotypical correlation)? Ex. do they often share the same GO process?

Minor comments:

When computing the genetic correlation of transcripts (page 11), what are the covariates in b_i and b_j ? Have the authors corrected for hidden confounders such as PEER factors?

Section "Identification of shared eSNPs": are the results in Figure 5 (page 5) obtained from the CAGE data or LIFE-Heart study?

The citation of CAGE study, Lloyd-Jones et al, 2016: no journal name or detailed references.

The order of the Figures seem arbitrary. Ex. Figure 5 is discussed in the text before Figures 2-4.

Reviewer #3 (Remarks to the Author):

This manuscript discusses the interpretation of genetic co-variation and co-localization of genetic regulation of gene expression in human cohorts. I was not clear on the main scientific findings of this paper, and how these findings differ from the existing work -- in particular, the existing work was not referenced and compared appropriately, and the methods and data were not described in a way that the results were interetable.

Major comments:

- what is the main take-home message of this paper, scientifically? How is this message different than existing work in this area (much of which was uncited)?
- related: what is the bigger picture of identifying "genetic correlation" and separating this from environment? what are the assumptions of these models, and how are the biases in the data controlled in the interpretation of the results?
- Why was this analysis not performed in RNA-seq data, instead of data from outdated microarray technology?
- p-values are meaningless without tests names; anything below 2.2×10^{-16} may be a rounding error, and should be truncated.
- bivariate GREML is far from the only method to do this estimation. All of the colocalization methods, for example, plus network methods and multivariate SNP association will ultimately include much of this information. Later in intro, the list of methods from authors of this paper are included -- please be more inclusive (to be clear, I'm not fishing here. I have not worked on this problem).

- "Identification of shared eSNPs" - this subsection is unclear. What is an eSNP and a probe? I thought a probe was for gene expression, not SNPs? How is this considered validation of the previous data?

- the network of co-regulated genes should be validated using existing pathways.

- I have looked at the Hi-C data for cross-chromosomal connections, and I think that there are a number of issues that confound the analysis, namely, 1) tissue type 2) resolution is low for cross-chromosomal contact; 3) background is incredibly noisy, making estimates difficult to evaluate

- how are the networks created? are they directed as described in the abstract? This needs to be specified in the Results section.

Minor comments:

In abstract (note: i went through the abstract very carefully, and pointed out all of the places where the writing was unclear or assumed too much from the reader. this type of text is pervasive throughout the manuscript. Please go carefully through each sentence to correct these issues, but I will stop reading this carefully after the abstract.)

- "Using a bivariate GREML approach" -- Please assume the reader has no idea what this is.

- "on different chromosomes to one another" remove "to one another"

- Hi-C validation of the different chromosome interactions?

- What is a "probe" and an "eSNP"

- what are "highly interconnected transcription factors"? do you mean shared, open transcription factor binding sites local to each gene? the p-value here is meaningless, because the null test was not stated.

- what is "enrichment of genetically correlated transcripts pairs and regions of chromatin interaction" -- enrichment of X and Y does not make the statistical test clear.

- I'm already skeptical of "the direction of shared genetic control" given what has been said before -- correlation does not mean causation, so it must be a different type of approach.

Introduction:

- "The majority of eQTL studies to date have analyzed associations between SNPs and individual transcripts" -- while this is certainly true, it is being used to imply that there are very few bivariate or multivariate SNP-transcript papers, which is absolutely not true. Should cite the many papers that discuss methods and applications of these methods to all of the existing eQTL data.

- "variants on the same, or different, chromosomes (cis/trans)" - this is sloppy. needs a real definition, and a community accepted one.

- "Suppose that the variation in the transcription factor's expression levels is entirely due to a single genetic variant." unless that SNP is coding and leads to LoF, TFs rarely have a large effect eQTL according to the recent literature.

- "A positive genetic correlation between two transcripts implies that, on average, shared genetic variants have an allelic effect in the same direction." -- also not true. could be shared

environmental effects with no SNP interactions.

- It is essential to check that the pairs of genes in the 506 and in the 14,991 do not cross-map or cross-hybridize. I would recommend a blast search on each pair to see if they have any shared sequences.

Results:

- Before diving into the results, a clear description of the data (platforms, genotypes, imputation, quantification, controlling for known covariates such as batch, etc.) needs to exist, as well as a clear description of the statistical approach to compute "genetic correlations".

- GCTA software needs a citation.

- what does 2,469 probes mean? are these microarray data?

- what does "converged results" mean? This concerns me. Presumably this means that for the other probe pairs, the model did not converge. This is a strange criterion and leads to presumably a biased subset of results. PDF is folded so I can't read the important line there.

- "significant proportion (36.6%)" -- significant according to what statistical test and null hypothesis? what is the p-value?

- "Probe pairs with both high r^P and r^G values likely represent members of networks with shared genetic regulatory mechanisms." -- why?

- "bias toward positive phenotypic and genetic correlations" -- again, quantify with a statistical test and a p-value.

- Same question about the data from the LIFE-Heart study: what cell type, array type, genotype assay, association method, quantification, correction was used? Describe the data basis.

- Bonferroni should be consistently capitalized. "Bonferroni significance threshold" isn't meaningful.

- "shared genetic control of transcription exists throughout the genome" disagree. could it be that the genetic control is local to one gene, which has global effects?

- "clear shift towards positive r^G " needs to be quantified with a statistical test and p-value.

- why did you use K562 cell lines?

Discussion:

- This section reads more like a subsection of specific examples in the Results than a discussion. Please revise, moving (previously undiscussed) results into the Results section.

- terminology (i.e., trans/trans shared eSNP) should not be introduced in the discussion.

Methods:

- many of the details in the methods are important to move to the Results.

- many of the methods, particularly the equations, are not well-described, specifically the meaning (and intuition behind) the specific parameters.

Response to the reviewer comments

Reviewer #2 (Remarks to the Author):

The manuscript by Powell and colleagues studied the interesting problem of genetic correlation (shared genetic basis) of gene expression traits. Comparing with the typical eQTL analysis, this is a very novel idea and can potentially provide more information of the underlying gene regulatory networks, not available from the eQTL perspective. The authors were able to identify tens of thousands of genetically correlate transcript pairs, and characterized the nature and pattern of such correlation, e.g. many of the cis-pairs (close in chromosome locations) correspond to chromatin interactions identified in Hi-C. The paper should be of broad interest to readers studying eQTL and gene regulation.

We would like to thank the reviewer for taking the time to review our work and for providing helpful comments and suggestions. We have addressed each comment, and provided a point-by-point response to each below.

Major comments:

Replication of genetic correlations in an independent data: The authors apparently have two datasets, their own eQTL data (CAGE?) and LIFE-Heart study. A natural question is whether they can replicate their r_G obtained in CAGE, in the second dataset. The authors seem to do something along the line of replication, when they described their study of shared eSNPs in the section "Identification of shared eSNPs" of Results, but a more direct way is just to compare r_G from two studies.

We agree wholeheartedly with the assessment made here regarding the possibility of replication in the LIFE-Heart study. Indeed, this strategy would have been preferable to us. However, unfortunately we were prohibited from applying the method required to estimate genetic correlations in population-level data due to the ethics and data access agreement of the LIFE-Heart study. Specifically, the problem lays in the requirement that relatedness needs to be calculated between pairs of individuals, using an Identical-By-State (IBS) method. This method will identify individuals who share a larger fraction of their genomes as IBS, and thus are considered related. This is a not uncommon phenomenon and will indirectly identify related individuals who previously had no knowledge of their relationship to one another. The ethics statements and data access agreement of the LIFE-Heart study

precluded analysis that would identify consanguineous relationships, false paternity and unknown relationships.

In light of this, we designed the strategy to assess an element of validation by identifying eQTL in the LIFE-Heart cohort that have significant associations with both transcripts in an rG pair. While we acknowledge that this is not the same as a complete replication of rG, it does benefit from the ability to demonstrate that specific loci can be identified in an independent cohort with shared effects. This is important as the rG estimates represent the averaged effects of all loci with effects on the two transcripts, and thus give no information on specific loci that underlie the observations. Our analysis of the shared eQTL effects presents evidence for this. We considered this analogous to the differences between estimating heritability for a phenotype and mapping loci for that phenotype. One could be considered an indirect replication or validation of the other, but also provides additional new knowledge.

Identification of shared eSNPs: in this section of Results, there are several issues about the analysis in the first paragraph. First, Figure S3 is a bit strange. The labels in S3A do not match the text, e.g. there is an arrow from eSNP_i to eSNP_{j,i}, but based on the text, it seems that it should be eSNP_i to eSNP_{i,j}. Second, there is no explanation of Figure S3B and C in the text. Why did the authors do this? Lastly, a natural question to ask here is that, given eSNP_i (the strongest eSNP of probe i), how often it is also associated, at some appropriate threshold, with a genetically correlated transcript? How does this proportion compare with random expectation?

We apologize for the confusion caused with Figure S3 (now S4) and naming convention of eSNPs and probes in the text. We have altered the suffixes for the eSNP-probe relationships in the text (lines 375-383), updated the figure and amended the figure legend. The updated version of Figure S4 more clearly describes the underlying approach taken with our analysis.

In the end of the first paragraph of page 5, the authors reported 100% replication. As I understood, among 934 eSNPs, about half represent trans-eQTL. This result is very striking. For all the trans-eQTL studies in literature, the reviewer's impression is that often trans-eQTL are tissue-specific and have low rates of replication. Presumably the 100% replication of trans-eQTL is to some extent due to the fact that these eQTL are associated with at least a pair of probes, thus more likely to be real eQTL. Assuming this is indeed the case, the authors should somehow highlight

the results and perhaps investigate a bit more to address a more general point/hypothesis: that large effect trans-eQTL, associated with multiple transcripts, can often be replicated.

We thank the reviewer for highlighting an important result in our manuscript that we did not adequately acknowledge. The prevalence of large-effect *trans*-eQTL that are associated with multiple transcripts in our data, and their replication is a striking result. To further investigate this, we obtained data for significant *trans*-eQTL from the Westra *et al.* (2013) study located at: (<http://genenetwork.nl/bloodeqtlbrowser/>). Within this dataset, each eSNP has a study-wide z-score (overall z-score) and also the z-scores for each of the individual cohorts that comprised the study. To determine the extent of replication for each *trans*-eQTL, we calculated the percentage of individual cohorts in which each SNP was replicated, and then plotted this against the absolute overall z-score (study-wide). We found that those *trans*-eQTLs with a large effect (overall z-score) also had a high rate of replication across multiple studies. (see Supplementary Figure 7). Furthermore, we show that the more transcripts that a *trans*-eQTL is associated with, the higher the replication rate.

We have edited the results section (lines 114-118) to include this update, and we have also amended the text for our initial finding, which showed that shared *trans*-eSNPs associated with multiple transcripts and were completely replicated in the CAGE dataset (lines 111-114).

The section "Genetically correlated transcripts are enriched in regions of interacting chromatin": there seems a clear enrichment of probe pairs that correspond to interacting pairs in Hi-C. However, the reviewer wonders whether this is really due to chromatin interactions. One expects that the regulatory loci (eQTL) interact with their targets, but we do not necessarily expect that two transcripts form chromatin interactions with each other. So the results seem somewhat puzzling here. Could the enrichment here be simply due to that the correlated pairs are often close to each other? Have the authors properly control for distance when obtaining the null distribution?

We agree with the reviewer's comment here. The Hi-C dataset was stringently filtered (Lan *et al.* 2012) to remove self-ligation events, proximate ligation events and random ligations. After filtering, 75% of interactions were within 1Mb, 20% > 1Mb and 5% inter-chromosomal. Since the majority of observed interactions are moderately close to each other, it is fair to assume many rG pairs that overlap with an interaction loci are in close proximity. However, in our analysis, we removed any

instance where both transcripts were within the same window region surrounding an observed Hi-C interaction. To increase confidence that the enrichments are not purely due to rG transcripts in close proximity, we performed our analysis and permutations with different distance parameters. As is shown in Supplementary Figure 7, larger distances have a substantially higher number of overlapping interactions, indicating that close rG pairs are not likely the source of enrichment.

We have added new text to the Discussion section (lines 301-309) clarifying that the mechanisms we have tested are likely part of a complex and multi-faceted regulatory system.

Network connectivity analysis at the end of page 7: is the analysis based on probes or transcripts? If it is probe-based, then the estimated connectivity may be inflated, as a single correlated transcript/gene pair may generate many probe pairs. Has this been properly addressed? The authors said that 2,317/2,468 probes have more than 113 connections at $r_G > 0.48$. What does the result say here - what's the biological significance? What do we expect by chance?

We apologize that the methods we used in this section were unclear. The analysis is based on transcripts that are measured using the Illumina HT-12v4 expression array, comprising of coding- and non-coding transcripts. In general, one probe maps to one transcript, however splice variants are also covered by the array design, allowing multiple transcript isoforms to be distinguished. It is possible that this results in inflation of the estimated connectivity. We asked whether removing a subset of probes that represent multiple transcripts for the same gene would alter the connectivity. In our test, we removed 50% of probes, assuming that each gene had two probes (i.e. two transcripts). We observed a proportional reduction in expected connection that may be attributable to inflated connectivity. However, we have shown in our data, using ZNF536 as an example, that transcript isoforms can form independent networks with limited crossover, or shared network genes. This supports the notion that multiple probes per gene that identify individual transcript isoforms are of biological importance, and may be involved in different biological pathways.

We asked how many transcripts would have rG pairs ≥ 0.48 (which corresponds to 2SD from the mean rG under the null), and then determined how many connections there were for each transcript. To ensure the number of observed connections was not due to chance, we calculated the expected number of connections as follows: using the percentage for 2 standard deviations based on a normal distribution (95.4%) and the number of unique probes in our data (2468), we used the equation:

$$(1 - 0.954) * 2468 = 113.528$$

This value was considered the number of connections expected at 2SD by chance, and those transcripts with connectivity greater than 113 were considered 'highly connected'. The biological significance of this is that a large number of transcripts have high numbers of rG pairs greater than expected by chance, and are therefore more likely to be at the hub of a network of genes/transcripts with shared genetic control.

We have edited the Results section to more clearly explain the biological significance of the network connections on lines 222-232.

The section "Highly connected transcripts are enriched for transcription factors": the description is not clear. If I understand correctly, the authors found that 229/234 TFs have more than 113 connections. To obtain the p-value of 1.14E-3, what was this compared against, 2317/2468?

We were interested in the following hypothesis: 'Transcription Factor (TF) genes are enriched for high genetic correlations compared to non-TF genes'. To address this, we first identified the number of global unique genes (2,330), the global expressed TFs (234), the observed TFs (229) and the number of unique gene IDs above the standard deviation threshold (2,193; 2SD: $|rG| \geq 0.48$). We used a hyper-geometric test to estimate the significance of the observed enrichment. We have clarified the description and values used to obtain our results for this section at lines 236-247.

In Table 2: h^2 seem really high, for some probes, h^2 is close to 1. Most estimates of expression heritability seem to be on the lower end, e.g. see Wright et al, Nature Genetics, 2014. Why is this the case? If the authors estimate h^2 from their genes, does the distribution agree with the distribution reported in other papers?

We only performed analyses on transcripts with heritability greater than 0.25, and thus represents a truncated distribution. However, the distribution of h^2 for all transcripts is in line with that published by Wright *et al.*, and is detailed in Lloyd-Jones *et al.* 2017. As the SE of the Rg estimate, and thus significance, is partly a function of h^2 , we have an enrichment of high heritability probes in the top 20 as given in Table 2.

It would be helpful if the authors can do more analysis and validation on the predicted transcript pairs. One obvious question is that are these pairs tend to be more functionally related than expected by chance (perhaps better, expected based on phenotypical correlation)? Ex. do they often share the same GO process?

We appreciate the suggestion to investigate whether our genetically correlated transcripts share biological function with one another. To test this, we performed a Gene Ontology (GO) enrichment analysis for the top 50 networks identified by genetic correlation analysis (those listed in figure 4). We compared this against a background of all transcripts. The top associated GO terms for each network are included in Supplementary Table 7. We have edited the results (lines 203-210) and methods (lines 361-366) sections to include this new analysis.

Minor comments:

When computing the genetic correlation of transcripts (page 11), what are the covariates in b_i and b_j ? Have the authors corrected for hidden confounders such as PEER factors?

We apologize. b_i and b_j are typos from an earlier draft and have now been removed from the manuscript. The CAGE data was normalized for both known and hidden confounders using the PEER method. Please see Lloyd-Jones *et al.* (2017) for details on the normalization strategy.

Section "Identification of shared eSNPs": are the results in Figure 5 (page 5) obtained from the CAGE data or LIFE-Heart study?

The shared eSNPs are from the LIFE-Heart Study. This cohort was used to ensure that genetic variants with shared effects on a pair of transcripts were identified in an independent cohort to the one used to estimate the genetic correlations between pairs of transcripts. We have now edited the text (see lines 97-109) to clarify the origin of the shared eSNPs.

The citation of CAGE study, Lloyd-Jones et al, 2016: no journal name or detailed references.

The Lloyd-Jones *et al.* manuscript is due to appear in the February 2017 issue of the American Journal of Human Genetics. We have updated the reference with the link

to the AJHG online version of the paper and will update the reference in full once issue and page numbers are available.

The order of the Figures seem arbitrary. Ex. Figure 5 is discussed in the text before Figures 2-4.

We have corrected the order of the Figures and Tables to match the manuscript text.

Reviewer #3 (Remarks to the Author):

This manuscript discusses the interpretation of genetic co-variation and co-localization of genetic regulation of gene expression in human cohorts. I was not clear on the main scientific findings of this paper, and how these findings differ from the existing work -- in particular, the existing work was not referenced and compared appropriately, and the methods and data were not described in a way that the results were interetable.

We would like to thank the referee for taking the time to review our manuscript, and for helpful comments, particularly around the lack of clarity and main scientific findings. We have edited the manuscript throughout, with particular emphasis on making the scientific message and interpretation of the results as clear as possible.

Major comments:

What is the main take-home message of this paper, scientifically? How is this message different than existing work in this area (much of which was uncited)?

We have now amended the text through to help present the concepts and take-home message more clearly. We have also incorporated a more comprehensive evaluation of the literature and outlined where our approaches have resolved previous problems and limitations.

Related: what is the bigger picture of identifying "genetic correlation" and separating this from environment? What are the assumptions of these models, and how are the biases in the data controlled in the interpretation of the results?

We appreciate that our manuscript did not accurately describe the 'bigger picture' of our work. We have now tried to address this with edits throughout, particularly the abstract and introduction (lines 6-8, 26-42). We have also changed the title to now

read '*Genetic correlations reveal the shared genetic architecture of*

Specifically, we wanted to understand the factors that lead to the observation that many transcripts are known to be co-expressed. We know that this observation is a combination of shared genetic and shared environmental factors. However, we did not know what was the extent of the genetic compared to non-genetic shared effects. Identifying the shared genetic effects also allows us to identify specific loci that have associated effects on pairs of transcripts, and identify networks and functional mechanisms for co-expression. Knowledge on this shared genetic architecture is also important in the context of human disease genomics, as much of the genetic susceptibility for human diseases are thought to act through changes in gene regulation. To date, most of this knowledge is based on univariate (mainly *cis*-)eQTL overlap with disease GWAS loci. We have now amended the text to

Why was this analysis not performed in RNA-seq data, instead of data from outdated microarray technology?

We agree that RNA-sequence would have been a preferable platform for us to obtain data for this research. However, one practical issue is that to accurately estimate genetic correlations between expressed transcripts, a large cohort is required and must contain both genotype and expression data for each individual. Unfortunately, we do not have access to this type of data from a cohort of sufficient size to enable us to accurately estimate genetic corrections. The expression array data that we used was normalized and prepared for analysis using robust and modern statistical techniques, which are outlined in the methodology of Lloyd-Jones *et al.*, 2017.

The CAGE dataset, from the Consortium for the Architecture of Gene Expression (CAGE), and currently represents one of the largest cohorts containing genotype and gene expression data for more than a total of 2,700 individuals (Lloyd-Jones *et al.*, 2017), of which 1,748 are unrelated individuals of European ancestry used in the current work. While the microarray platform is no longer the 'state-of-the-art', we believe that the data remains high quality and suitable for expression studies across many individuals.

P-values are meaningless without tests names; anything below 2.2×10^{-16} may be a rounding error, and should be truncated.

We agree with the reviewer regarding test names, which have now been included for instances where a *p*-value is given. However, we are confused about the comment about *p*-values below 2.2×10^{-16} . To our knowledge *p*-values can be

computed reliably (using IEEE double precision floats) as small as $\sim 10^{-303}$. Certain functions in R have a read out limit of 2.2×10^{-16} , but this is due to their default floating-point operation being `double.eps`.

Bivariate GREML is far from the only method to do this estimation. All of the colocalization methods, for example, plus network methods and multivariate SNP association will ultimately include much of this information. Later in intro, the list of methods from authors of this paper are included -- please be more inclusive (to be clear, I'm not fishing here. I have not worked on this problem).

We appreciate this point and have included additional text, particularly around the literature showing multivariate eQTLs and other methods for estimating genetic correlations (lines 26-33 and 297-300). Estimating genetic correlations provides a way to sum the effects of all genetic variants that have a shared effect on a pair of transcripts, and thus have some distinction to models identifying specific loci.

"Identification of shared eSNPs" - this subsection is unclear. What is an eSNP and a probe? I thought a probe was for gene expression, not SNPs? How is this considered validation of the previous data?

We thank the reviewer for highlighting the lack of clarity with respect to the description of the 'shared eSNP' analysis. The term 'eSNP' is commonly used as the descriptor for the SNP that has been shown to have a significant effect on the expression of a transcript in an eQTL study. The probes reflect the transcript expression, and as such, an expression SNP (eSNP) is a SNP significantly associated with the expression levels of one (or more) transcript. We have edited the manuscript at lines 97-101 and 375-381 with a clearer definition of eSNPs and a better explanation of the shared regulatory effects we were looking for.

Our intention was not to validate findings from a previous dataset, but to identify the type of genetic regulatory effects measured when estimating r_G . By overlaying independent eQTL data (LIFE-Heart) with r_G data, we were able to verify specific loci with allelic effects on both transcripts, which underlie our observed genetic correlations. We apologize for the lack of clarity surrounding this. We have edited the text accordingly (see lines 107-109).

The network of co-regulated genes should be validated using existing pathways.

We feel that it is important to note that our analysis has identified networks of genes based upon a high degree of shared genetic variance underlying their transcriptional variation, and not based cellular functions or processes. We sought to validate these networks by showing that there are individual genetic variants that are associated with the expression levels of multiple transcripts, and that these associations would not be expected by chance. We performed this analysis in completely independent data (LIFE-Heart study).

Nevertheless, we appreciate the suggestion that our networks based upon genetic covariance may reveal interesting observations on biological functions. As such, we have now performed a Gene Ontology (GO) enrichment analysis for the top 50 networks identified by genetic correlation analysis. The top associated GO terms for each network are included in Supplementary Table 7. We have edited the results (lines 203-210) and methods (lines 361-366) sections to include this new analysis.

I have looked at the Hi-C data for cross-chromosomal connections, and I think that there are a number of issues that confound the analysis, namely, 1) tissue type 2) resolution is low for cross-chromosomal contact; 3) background is incredibly noisy, making estimates difficult to evaluate.

We thank the reviewer for their insightful comments. Our response attempts to address each question individually.

1. Tissue-type:

We acknowledge the difference in tissue types between our study (blood) and the Hi-C data (K562 erythroleukemic cells). We have now included results using Hi-C data for a lymphoblastoid cell line (LCLs: GM06990). This includes the test for enrichment of significant rG pairs of transcripts in intra-chromosomal contacts using the LCL and K562 data and intra-chromosomal rG probe pairs from the Bonferroni and FDR subsets. We have also included a K562 inter-chromosomal analysis for the inter-chromosomal rG probe pair from the same Bonferroni and FDR subsets. These are shown in Figure 4 and Supplementary Figure 7, and the text edits in the methods and results sections (lines 413-431 and 162-179). While we acknowledge that LCLs are not ideal, from an extensive literature search, they were the closest cell type for which publically available Hi-C results were obtainable. However, only intra-chromosomal contacts were published.

From a biological perspective, K562 cells are considered undifferentiated progenitors that can differentiate into mature cell types such as monocytes. The genome of K562 cells is rearranged compared to GM06990/GM12878 cells, which could inflate the

proportion of inter-chromosomal contacts based on fusion chromosomes that are actually intra-chromosomal for the normal genome. However, Lan et al (2012) found that, except for the region of the Philadelphia chromosome (t(9;22)(q34;q11)), only two regions of chromosomal translocation contributed to inter-chromosomal contacts. This indicates the majority of inter-chromosomal contacts in K562 cells are not due to genomic rearrangements. As many chromosomal interactions are tissue-specific and temporally dynamic, we appreciate that the use of K562 cells may obscure a proportion of real interactions occurring in blood. However, our study was not designed to identify all possible interactions, but merely to show that genetically correlated transcripts may undergo regulation via chromosomal looping or inter-chromosomal contacts. With this in mind, we feel the use of K562 cells in this section of the analysis is justified, and supported by results from the LCL analysis.

2. Resolution:

We also acknowledge that the resolution of the Hi-C data (1Mb bins) is low and that this may reduce the number of inter-chromosomal contacts that can be observed. To our knowledge, an equivalent, tissue-similar dataset measured with a larger bin resolution is not available. In addition, the stringent filtering of the K562 data, performed by Lan *et al.* 2012, eliminates a high number of contacts purported to be false positives.

3. Background noise:

Stringent QC and filtering as outlined by Heinz *et al.* (2009) and Lan *et al.* (2012) was applied to minimise the presence of random ligations and other sources of experimental noise. Finally, with respect to background noise of the rG data, we appreciate that the variance of each rG estimate is moderate, however, rG measured with sufficient accuracy that we could completely replicate the effects of eSNPs on expression probes in an independent study, and capture known interactions between several transcripts located on different chromosomes.

How are the networks created? are they directed as described in the abstract? This needs to be specified in the Results section.

The graphical constructions of networks were created using the Fruchterman-Reingold algorithm implemented in the 'igraph' software. In our analyses, the graph networks are not directed, in that the edges are bidirectional and thus do not denote any direction of causal effect. However, we coloured the edges based on a positive or negative rG estimates between the pairs of transcripts. This provides information on the average allelic direction for genetic loci underlying the genetic correlations. We have edited the abstract to avoid confusion over the use of the term directed,

and edited the results (lines 181-186 and 193-196) and methods (401-403) sections to clarify the methods used.

Minor comments:

In abstract (note: i went through the abstract very carefully, and pointed out all of the places where the writing was unclear or assumed too much from the reader. this type of text is pervasive throughout the manuscript. Please go carefully through each sentence to correct these issues, but I will stop reading this carefully after the abstract.)

We have now edited the manuscript throughout. Below is a response to each of the specific comments.

"Using a bivariate GREML approach" -- Please assume the reader has no idea what this is.

We have removed this phrase from the abstract.

"on different chromosomes to one another" remove "to one another"

This has been edited to now read '...on different chromosomes' as suggested.

Hi-C validation of the different chromosome interactions?

We have removed this sentence from the abstract.

What is a "probe" and an "eSNP"

Definitions for both probe and eSNP have been given in the text. See lines 325-326 and 98-99.

what are "highly interconnected transcription factors"? do you mean shared, open transcription factor binding sites local to each gene? the p-value here is meaningless, because the null test was not stated.

We are sorry for the confusion here. In the context of this paper, highly interconnected transcription factors are those with the highest number of rG pairings above a statistically determined threshold of 3 standard deviations from the mean. The definition of highly interconnected transcription factors is stated in the text and methods on lines 233-247. We have removed the p -value from the abstract and edited the text to provide a clearer definition.

what is "enrichment of genetically correlated transcripts pairs and regions of chromatin interaction" -- enrichment of X and Y does not make the statistical test clear.

We agree that the statistical test is unclear. The 'enrichment' statement has been clarified in the text (see abstract and lines 161-163). We have also clarified the statistical tests used to demonstrate these results in lines 169-179.

I'm already skeptical of "the direction of shared genetic control" given what has been said before -- correlation does not mean causation, so it must be a different type of approach.

The direction of shared genetic control does not imply correlation == causation. Rather is it the numerical sign of the rG estimate (i.e. positive or negative) as this denotes the relationship between the average allelic directions of any loci that have an effect on both transcripts. We have now included additional text in the introduction to clearly describe this property (lines 49-54).

Introduction:

"The majority of eQTL studies to date have analyzed associations between SNPs and individual transcripts" -- while this is certainly true, it is being used to imply that there are very few bivariate or multivariate SNP-transcript papers, which is absolutely not true. Should cite the many papers that discuss methods and applications of these methods to all of the existing eQTL data.

We have edited the introduction and discussion to include commentary on the literature of multivariate eQTL analyses and their relevance in the context of estimating total shared genetic control between transcripts. Please see lines 26-33 and 294-300.

"variants on the same, or different, chromosomes (cis/trans)" - this is sloppy. needs a real definition, and a community accepted one.

We acknowledge the lack of clarity in this context. We agree that a clearer definition is required and have altered the manuscript accordingly. Where '*cis*' or '*trans*' was used, we have changed the text to 'intra-chromosomal' (*cis*), or 'inter-chromosomal' (*trans*) throughout the manuscript where appropriate.

"Suppose that the variation in the transcription factor's expression levels is entirely due to a single genetic variant." unless that SNP is coding and leads to LoF, TFs rarely have a large effect eQTL according to the recent literature.

While we appreciate that this scenario is not expected to be common, we were simply using it as a hypothetical example to connect the statistical properties (and their observations) of genetic correlations and a (potential) biological mechanism. Nevertheless, there is precedent for this example (<http://www.biorxiv.org/content/early/2016/06/19/059873>). We have amended the text (lines 43-49) to clarify that this is purely a simplified illustrative example.

"A positive genetic correlation between two transcripts implies that, on average, shared genetic variants have an allelic effect in the same direction." -- also not true. could be shared environmental effects with no SNP interactions.

The relationship between the expression levels of two transcripts, their phenotypic correlation (r_P), is due to the combined influence of shared genetic (r_G) and environmental factors (r_E) (see Cheverud 1988, *Evolution* 42, 958-968 for details). Genetic correlations (r_G) are the component of r_P that are due to genetic factors only. Shared environmental effects are likely to be very common, but these factors would form the r_E . As the genetic correlation is a summed estimate of the effects of all variants, including those with small effects that are not identifiable through current eQTL studies, it is correct to state that 'A positive genetic correlation between two transcripts implies that, on average, shared genetic variants have an allelic effect in the same direction'.

It is essential to check that the pairs of genes in the 506 and in the 14,991 do not cross-map or cross-hybridize. I would recommend a blast search on each pair to

see if they have any shared sequences.

Yes, as part of our quality control of the original data we removed all probes that showed evidence of cross-hybridization. Details are given in the supporting material of Lloyd-Jones *et al.* 2017.

Results:

Before diving into the results, a clear description of the data (platforms, genotypes, imputation, quantification, controlling for known covariates such as batch, etc.) needs to exist, as well as a clear description of the statistical approach to compute "genetic correlations".

A complete description of the data analyzed in this study is available in the Lloyd-Jones *et al.* (2017) manuscript and supporting material. However, we have included brief details in the methods section (lines 344-360). The statistical methods (GREML) and the variation (bivariate GREML) used in our study to estimate genetic correlations are well-established methods in quantitative genetics, and numerous papers have been published with details of the GREML method (Lee *et al.* 2011 (AJHG); Yang *et al.* 2011 (Nat Genet); Yang *et al.* 2012 (Nat Genet)).

The computational implementation of GREML is part of the GCTA software we used, and was published in Yang *et al.* 2011 (American Journal of Human Genetics). Bivariate GREML methodology is described in Lee *et al.*, 2012 (Bioinformatics). We have cited these articles where appropriate and edited the text in the introduction and methods to provide a clearer description of the concepts behind this methodology.

GCTA software needs a citation.

We have now also included a citation for this in the results section.

what does 2,469 probes mean? are these microarray data?

The expression data used in this study is from the Illumina HT-12v4 expression microarray. The expression levels of transcripts are tagged by array probes. We have amended the text in the results and methods to explain the use of the term probes and have edited the manuscript throughout from probe(s) -> transcript(s) as a clearer definition. The transcripts that we analyzed were determined to have a

minimum estimated narrow-sense heritability (h^2) of 0.25 or greater. h^2 was estimated for each of the transcript measured on the HT-12v4 array in Lloyd-Jones *et al.* (2017), and of these 2,469 had $h^2 > 0.25$. We set a threshold of $h^2 > 0.25$ due to the limited power to detect rG for transcripts with $h^2 < 0.25$. This is detailed in the methods section (lines 322-342).

what does "converged results" mean? This concerns me. Presumably this means that for the other probe pairs, the model did not converge. This is a strange criterion and leads to presumably a biased subset of results. PDF is folded so I can't read the important line there.

‘Converged results’ refer to the resolved output of the REML analysis. When the true corrections (i.e. the observed phenotypic correlations) or estimates (such as rG and rE) are close to zero, the maximum likelihood algorithm will fluctuate substantially in the REML often causes a non-positive definite variance-covariance matrix. This means that the REML iteration cannot process further and returns a ‘non-converged’ result for a specific pair of probes. Essentially non-converged results are those rG estimates for which the REML algorithm cannot estimate the covariance parameters and this is a common property of the REML algorithm. In our analysis, non-converged results were shown to come from pairs of probes that had a phenotypic correlation close to zero. Thus, we do not expect these transcripts to have any true genetic correlations between them.

"significant proportion (36.6%)" -- significant according to what statistical test and null hypothesis? what is the p-value?

We have removed the term ‘significant proportion’. Please see lines 88-89.

"Probe pairs with both high r^P and r^G values likely represent members of networks with shared genetic regulatory mechanisms." -- why?

We have provided further explanation in the text. Please see lines 90-93.

"bias toward positive phenotypic and genetic correlations" -- again, quantify with a statistical test and a p-value.

This section has been edited to read “a higher proportion of positive phenotypic and

genetic correlations between transcript pairs” and now includes the values of the positive and negative groups on lines 86-87. In addition, we performed a Shapiro-Wilk test for normality using (i) a random normal distribution of 5,000 values, and (ii) a random sample of 5,000 rP values. We observed a significant deviation from a normal distribution for the rP values ($p < 2.2e-16$), with an increased density of positive values. This plot has been included in the supplementary materials (Figure S3).

Same question about the data from the LIFE-Heart study: what cell type, array type, genotype assay, association method, quantification, correction was used? Describe the data basis.

The LIFE-Heart study was published previously and referenced accordingly, and the experimental details give in detail in Kirsten *et al.* 2015. However, we have edited our manuscript to provide clearer details and referencing in the Methods section, under the LIFE-Heart eQTL subheading. Specifically, we state that the data was generated from peripheral blood monocytes using Illumina HT12 expression arrays, and the genotypes of each individual were imputed against the 1000 Genome (phase1 v3) reference panel. Additional details pertaining to the normalization, association method, quantification and correction are given in our manuscript and referenced to Kirsten *et al.* (2015).

Bonferroni should be consistently capitalized. "Bonferroni significance threshold" isn't meaningful.

We have clarified the Bonferroni thresholds throughout the manuscript, and have edited the text to include the calculations used to determine threshold values. Bonferroni has been consistently capitalized.

"shared genetic control of transcription exists throughout the genome" disagree. could it be that the genetic control is local to one gene, which has global effects?

We apologize for the confusion around this statement and the concept of shared genetic control. We wholeheartedly agree with the reviewer that genetic control could be local to a single gene that has global effects, and this would constitute one of many examples of shared genetic control. By ‘shared genetic control of transcription’, we mean that a genetic regulatory mechanism is common to set of transcripts that are genetically correlated. *i.e.* genetic variance that underlies some

of the observed expression variance is shared between two transcripts at two or more positions in the genome. In our manuscript we give three detailed examples of shared genetic control (Figure 2). The text in the manuscript has been edited throughout, and we believe the concept of shared genetic control is more accurately explained.

"clear shift towards positive r^2 " needs to be quantified with a statistical test and p-value.

We have edited this statement for clarity on lines 152-153. It now reads 'we observed an increase in the percentage of pairs with shared eSNPs at tails of the r^2 estimate distribution'.

why did you use K562 cell lines?

Please see our response to the previous, highly detailed comment regarding the Hi-C analysis.

Discussion:

This section reads more like a subsection of specific examples in the Results than a discussion. Please revise, moving (previously undiscussed) results into the Results section.

We have restructured the discussion throughout. This is inline with making the scientific message and interpretation of the results as clear as possible.

terminology (i.e., trans/trans shared eSNP) should not be introduced in the discussion.

The manuscript has been edited so that this terminology is introduced in the Results section where it is first described on line 128.

Methods:

many of the details in the methods are important to move to the Results.

Both methods and results have been edited accordingly.

many of the methods, particularly the equations, are not well-described, specifically the meaning (and intuition behind) the specific parameters.

The methods section has been edited throughout, with clearer descriptions of the meaning or reasons for use of specific methods. Where relevant we have referenced the original papers for each equation and method. Likewise, parameters that are specific to this analysis that are not previously published are defined in the Methods section where appropriate.

Reviewers' comments:

Reviewer #2 (Remarks to the Author):

The authors have addressed all my concerns. I have only one minor comment. In line 115, the authors said, "trans-eQTL with effects on multiple transcripts were consistently replicated". However, this result is not shown in Figure S5 (only the percent of replication vs. effect size, not the number of associated transcripts).

Reviewer #4 (Remarks to the Author):

The role of chromatin architecture in transcriptional regulation is a neglected area of research. This manuscript is an attempt to address this important subject. There are, however, some issues with the chromatin interaction analysis that needs to be addressed:

(1) Authors used the quantitative microarray data to demonstrate correlated level of expression of a number of genes in human peripheral blood. Microarray or RNA-Seq analysis measures steady state RNA level, which is influenced by transcription as well as the stability of the transcript. The GRO-Seq and NET-Seq analysis are therefore currently used for genomewide transcription analysis. Authors should acknowledge this in the 'Discussion' section.

(2) Authors used published Hi-C data obtained with K562 erythrocytic cells and lymphoblastoid cell line (Lieberman-Aiden et al. 2009; Lan et al. 2012). Chromatin interaction map obtained by Hi-C analysis vary from cell type to cell type and within a cell type under different set of conditions. Correlating Hi-C data from K562 and lymphoblastoid cells with RNA data from peripheral blood may lead to erroneous conclusions. Authors can perform CCC analysis for selected genes in their blood samples following the protocol published by Dekker lab. In the same batch of cells, they should perform nascent RNA analysis following the method published by John Lis lab (Core et al., 2008). They should then compare the chromatin interaction of just a few loci shown in Fig. 2 with their nascent transcript level. If the CCC analysis and RNA expression analysis for selected genes shown in Fig. 2 are the same as reported by authors in this paper, it will greatly improve the confidence in their data.

(3) Chromatin interaction data in Fig 4 was obtained at 500 kbp resolution. Authors can use the same Hi-C data (Lieberman-Aiden et al. 2009 and Lan et al. 2012) to see the interaction of pairs of selected loci exhibiting co-expression at a lower resolution (100-1000 bp resolution). Whether the loci exhibiting co-expression shown in Fig. 2 are physically interacting with each other and with eSNP can be easily determined. This will again greatly improve the confidence in their data.

Response to the reviewer comments

Reviewers' comments:

Reviewer #2 (Remarks to the Author):

The authors have addressed all my concerns. I have only one minor comment. In line 115, the authors said, "trans-eQTL with effects on multiple transcripts were consistently replicated". However, this result is not shown in Figure S5 (only the percent of replication vs. effect size, not the number of associated transcripts).

We have added new text (lines 118-119) to expand the interpretation of the result with the number of transcripts associated with replicating *trans*-eQTL. We are grateful to the reviewer for their time and thorough review.

Reviewer #4 (Remarks to the Author):

The role of chromatin architecture in transcriptional regulation is a neglected area of research. This manuscript is an attempt to address this important subject. There are, however, some issues with the chromatin interaction analysis that needs to be addressed:

(1) Authors used the quantitative microarray data to demonstrate correlated level of expression of a number of genes in human peripheral blood. Microarray or RNA-Seq analysis measures steady state RNA level, which is influenced by transcription as well as the stability of the transcript. The GRO-Seq and NET-Seq analysis are therefore currently used for genomewide transcription analysis. Authors should acknowledge this in the 'Discussion' section.

We thank the reviewer for bringing this important distinction to our attention. We have amended the text in the Discussion (lines 286-290) accordingly.

(2) Authors used published Hi-C data obtained with K562 erythroleukemic cells and lymphoblastoid cell line (Lieberman-Aiden et al. 2009; Lan et al. 2012). Chromatin interaction map obtained by Hi-C analysis vary from cell type to cell type and within a cell type under different set of conditions. Correlating Hi-C data from K562 and lymphoblastoid cells with RNA data from peripheral blood may lead to erroneous conclusions. Authors can perform CCC analysis for selected genes in their blood samples following the protocol published by Dekker lab. In the same batch of cells, they should perform nascent RNA analysis following the method published by John Lis lab (Core et al., 2008). They should then compare the chromatin interaction of just a few loci shown in Fig. 2 with their nascent transcript level. If the CCC analysis and RNA expression analysis for selected genes shown in Fig. 2 are the same as reported by authors in this paper, it will greatly improve the confidence in their data.

In our work, we used two cell lines whose expression closely represents that of cells in blood. We acknowledge there are limitations in using proxy cells to draw strong biological conclusions, however there is justification for our decision. The generation of Hi-C data relies on a large number of cells in order to detect interactions with sufficient power. This is a primary reason for performing CCC experiments on derived cells. To account for differences between cell types and reduce error, we compared data from two well-studied blood-derived cell lines that are frequently used as a proxy for blood-based analyses.

While we appreciate the suggestion to perform the CCC analysis for selected genes in our blood samples, and agree this would represent the ideal verification of our findings, we believe this is unfeasible for this manuscript for two reasons. (1) Logistically, it is not possible to obtain sufficient blood samples from CAGE-cohort individuals to test a selection of eSNPs using CCC methodology. Tissue samples are stored in 6 sites internationally. The CAGE consortium only provides ethical and data transfer agreements to share genotype and transcriptional data, not blood samples. (2) Performing the CCC analysis requires personnel with specialised skills and specific laboratory equipment and worked protocols. Unfortunately our labs do not have these expertise, and thus to perform the analyses would require considerable time and financial resources.

Regarding the second part of this comment, although confirming real interactions using CCC for selected genes would provide strong evidence of transcriptional regulation mediated by eSNPs, for several reasons, to undertake this assay in whole blood is not feasible. Firstly, the effect of an eSNP can be detected due to the increased statistical power of a large cohort, but the effect size of a given eSNP is likely to be small. Improving the detection of regulatory events that would normally be hidden is the underlying reason we leveraged genetic correlations across a large cohort, and is based on the premise that the correlation is based on the average effects of all SNPs. This means that, while we can narrow down pairs or groups of genetically correlated transcripts, additional functional information, such as eSNP-mediated mechanisms of regulation, requires the generation of discrete models for functional validation. Although we are in agreement with the reviewer regarding functional validation of eSNP-mediated genetic regulation, we believe this is beyond the scope of the study.

(3) Chromatin interaction data in Fig 4 was obtained at 500 kbp resolution. Authors can use the same Hi-C data (Lieberman-Aiden et al. 2009 and Lan et al. 2012) to see the interaction of pairs of selected loci exhibiting co-expression at a lower resolution (100-1000 bp resolution). Whether the loci exhibiting co-expression shown in Fig. 2 are physically interacting with each other and with eSNP can be easily determined. This will again greatly improve the confidence in their data.

We thank the reviewer for this constructive comment. We performed additional analyses using the LCL and K562 data to identify chromatin contacts located within 1000bp of an rG probe pair. For this we used the intra- and inter-chromosomal rG subsets with an FDR threshold of 0.05, which provided 971 and 14,020 pairs for analysis respectively. For intra-chromosomal interactions, we observed zero overlaps for both the LCL and K562 chromatin contacts in a 1000bp window. For inter-chromosomal interactions, such as those highlighted in Figure 2, we also observed zero overlaps using the K562 dataset. We additionally performed 1,000 permutations of random loci for each dataset and did not detect any random interactions in a 1000bp window.

While the determination of the eSNP associated with interacting loci would allow fine-grained dissection of such a regulatory mechanism, the size and stochastic nature of chromatin contacts in a given cell type and state make the probability of detecting a contact in such a small window low. In general, chromatin interactions are visualised in the hundreds of kilobases to megabase size, and require high numbers of valid loci pairs to properly capture the interaction with statistical robustness, inhibiting the precise determination of interacting eSNPs using Hi-C data.

In our study, we used chromatin interactions to highlight regions of the genome that could interact with genetically correlated/co-regulated transcripts, or bring two genomic regions encompassing two rG transcripts into close proximity. We used windows of different sizes surrounding an rG probe to detect those that overlapped with chromatin contact loci. In the main figure, we show an analysis window of 500kb, and our supplementary data (Figure S7) shows the same analyses in windows of 1Mb, 500kb, 250kb and 100kb. With decreasing window size, we observed fewer overlaps with Hi-C loci. By reducing the window size to 1000bp, we no longer observed any overlaps with Hi-C data, and we believe this is likely due to the low probability of a small region of the genome being observed interacting with another small region in a very large chromatin region.

We have added new text in the Results section (lines 176-182) to reflect these new findings.

REVIEWERS' COMMENTS:

Reviewer #4 (Remarks to the Author):

The authors have addressed all the issues raised in my previous communication.